# An ALYREF-MYCN coactivator complex drives neuroblastoma tumorigenesis through effects on USP3 and MYCN stability

Zsuzsanna Nagy[1,2], Janith A. Seneviratne [1], Maxwell Kanikevich[1], William Chang[1], Chelsea Mayoh [1,2], Pooja Venkat [1], Yanhua Du [3], Cizhong Jiang [3], Alice Salib[1], Jessica Koach[1], Daniel R. Carter [1,2,4], Rituparna Mittra[1], Tao Liu [1], Michael W. Parker [5,6], Belamy B. Cheung [1,2,3]✉ & Glenn M. Marshall [1,2,7]✉

To achieve the very high oncoprotein levels required to drive the malignant state cancer cells utilise the ubiquitin proteasome system to upregulate transcription factor levels. Here our analyses identify *ALYREF*, expressed from the most common genetic copy number variation in neuroblastoma, chromosome 17q21-ter gain as a key regulator of MYCN protein turnover. We show strong co-operativity between ALYREF and MYCN from transgenic models of neuroblastoma in vitro and in vivo. The two proteins form a nuclear coactivator complex which stimulates transcription of the ubiquitin specific peptidase 3, USP3. We show that increased USP3 levels reduce K-48- and K-63-linked ubiquitination of MYCN, thus driving up MYCN protein stability. In the MYCN-ALYREF-USP3 signal, ALYREF is required for MYCN effects on the malignant phenotype and that of USP3 on MYCN stability. This data defines a MYCN oncoprotein dependency state which provides a rationale for future pharmacological studies.

[1] Children's Cancer Institute Australia for Medical Research, Lowy Cancer Research Centre, UNSW, Sydney, NSW, Australia. [2] School of Women's and Children's Health, UNSW Sydney, Randwick, NSW, Australia. [3] School of Life Sciences and Technology, Tongji University, Shanghai, China. [4] School of Biomedical Engineering, University of Technology, Sydney, NSW, Australia. [5] Department of Biochemistry and Molecular Biology, Bio21 Molecular Science and Biotechnology Institute, The University of Melbourne, Parkville, VIC, Australia. [6] ACRF Rational Drug Discovery Centre, St. Vincent's Institute of Medical Research, Fitzroy, VIC, Australia. [7] Kids Cancer Centre, Sydney Children's Hospital, Randwick, NSW, Australia. ✉email: bcheung@ccia.unsw.edu.au; GMarshall@ccia.unsw.edu.au

MYCN is a short-lived transcription factor, tightly controlled by the ubiquitin-proteasome system. Oncogenic stabilization of MYCN promotes target gene recognition and transcriptional programs involved in almost every aspect of tumorigenesis[1]. Amplification of the *MYCN* oncogene from 2p24 occurs in 25% of neuroblastoma patients, and is a well-recognized marker of tumor aggressiveness[2,3]. *MYCN*-amplification in primary neuroblastoma tissues shows strong statistical association with 17q21-ter gain[4,5], the most frequent genetic aberration in neuroblastoma at an incidence of 38–65%[4,6–11]. Genes identified within this chromosomal region such as *insulin growth factor 2 binding protein 1*[12], *survivin*[13], and *JMJD6*[14] have been reported to promote neuroblastoma cell survival through a wide range of mechanisms, however, the mechanistic relationship between MYCN and 17q21-ter is still not understood. As there are no specific inhibitors currently available for high-risk neuroblastoma patients with these features, there is an urgent need to identify new targetable molecular vulnerabilities.

The *ALYREF* gene is sited at chromosome 17q25.3, and codes for a ubiquitously expressed nuclear chaperone protein that controls many biological processes. ALYREF is a bZIP enhancing factor[15] that regulates DNA binding and transcriptional activity of RUNX1B and c-Myb[16], LEF-1 and AML[17], and E2F2[18]. ALYREF can bind to RNA polymerase II to assist in transcriptional elongation[19]. ALYREF is also a component of the TREX protein complex regulating nuclear export of mRNAs[20] and splicing[21]. *ALYREF* expression is dysregulated in several human cancers[22,23]. Suppression of ALYREF expression resulted in decreased cell proliferation[24], and the migratory capacity of oral squamous carcinoma cells[23]. Despite considerable study, it is not yet clear how the expression of ALYREF is increased or how elevated levels of ALYREF promote tumorigenesis.

In this study, we describe ALYREF as a key regulator of MYCN turnover and neuroblastoma tumorigenesis. Bioinformatic analyses demonstrated that elevated *ALYREF* expression associated with 17q21-ter gain, *MYCN*-amplification and was a predictor of poor patient survival. Mechanistic experiments revealed that MYCN was dependent on ALYREF to drive neuroblastoma cell proliferation and the two proteins acted in a positive interconnected regulatory pathway that was essential for sustaining mutual, oncogenic expression levels. ALYREF contributed to neuroblastoma cell proliferation by blocking MYCN degradation through direct transcriptional upregulation of *USP3*, a deubiquitinating enzyme. Genetic suppression of ALYREF expression resulted in reduced neuroblastoma cell proliferation in vitro, and delayed neuroblastoma growth in mice. These findings provide evidence for future pharmacological studies aimed at the development of ALYREF inhibitors as selective therapeutics for neuroblastoma with 17q21-ter gain and *MYCN*-amplification.

## Results

**High *ALYREF* expression associates with 17q21-ter gain, *MYCN*-amplification, and poor patient prognosis in neuroblastoma.** To identify molecular vulnerabilities in the context of 17q21-ter gains we first used a bioinformatic approach and analyzed the US TARGET neuroblastoma tumor cohort of 135 patients with Whole Genome Sequencing (WGS) data[25]. Analysis of the dataset showed a high frequency of gain beyond 17q21.31 (Fig. 1a) in neuroblastoma tissues. Kaplan–Meier analysis of overall survival probability of patients in the same tumor cohort ($n = 135$) showed that 17q21-ter gain (gain subgroup) was strongly predictive of poor patient survival when compared to patients without 17q21-ter gain (diploid subgroup; Fig. 1b). Amplification of the *MYCN* oncogene and consequent overexpression of the MYCN oncoprotein occur in 25% of

neuroblastoma patients and correlate with poor patient survival[2,3]. Further stratification of Kaplan–Meier survival analysis into four subgroups based on both 17q21-ter gain (gain or diploid) and *MYCN*-amplification status (amplified; MA, non-amplified; MNA) showed that patients with 17q21-ter gain and *MYCN*-amplification (gain-MA) had the worst survival probability among the four subgroups with significantly lower survival rates when compared to 17q21-ter gain (gain-MNA) alone or without either aberration (diploid-MNA) (Fig. 1c). We further utilized RNA-seq data from the same TARGET neuroblastoma cohort to analyze 1044 transcripts from the 17q21-ter locus for all of the following five criteria (Fig. 1d): differential gene expression in (1) 17q21-ter-gain patients (Supplementary Data 1) and (2) *MYCN*-amplified patients (Supplementary Table 1), correlation with (3) gene copy number (Supplementary Data 2) and (4) *MYCN* expression (Supplementary Table 2), as well as (5) association with poor neuroblastoma patient outcome (Supplementary Table 3). *ALYREF* was the only gene to pass all five criteria among the 1044 17q21-ter genes (Fig. 1e). Following transcriptome-wide differential gene expression analyses we observed that *ALYREF* was differentially expressed in patients with either 17q21-ter gain (Fig. 1f) or *MYCN*-amplification (Fig. 1g). We also observed genes previously associated with 17q21-ter gain in neuroblastoma, such as TBX2[26], PPMD1[27], JMJD6[14], and BIRC5[28] being differentially expressed in 17q21-ter gain patients (Supplementary Fig. 1a), whereas *MYCN* mRNA expression associated with *MYCN*-amplification (Supplementary Fig. 1b) but not 17q21-ter gain (Supplementary Fig. 1a). Cox proportional hazards (CoxPH) and Kaplan–Meier analyses of overall survival probability of patients in the TARGET cohort ($n = 154$) showed that high *ALYREF* expression was strongly predictive of poor patient survival (Fig. 1h). We obtained similar results from the larger RNA-seq SEQC ($n = 498$) neuroblastoma patient cohort[29–32] (Supplementary Fig. 1c). High *ALYREF* expression in this cohort also correlated with other predictors of poor prognosis in the disease, such as (1) *MYCN*-amplification, (2) advanced clinical stage (3/4), and (3) older age of the patient at diagnosis (Supplementary Fig. 1d). Multivariate CoxPH analysis for both overall and event-free (Supplementary Fig. 1e) survival in the SEQC cohort ($n = 498$) showed that high *ALYREF* expression retained independent prognostic significance, when compared with other prognostic factors such as patient age, clinical stage, and *MYCN*-amplification. Further additive CoxPH regression analysis for both overall and event-free (Supplementary Fig. 1f) survival showed that clinical stage and *MYCN*-amplification had the greatest impact on the prognostic significance of *ALYREF* expression. These data suggest that high *ALYREF* expression strongly predicts poor patient prognosis and that MYCN and ALYREF may co-operate as tumorigenic factors in neuroblastoma. We saw significant positive correlations between *ALYREF* and *MYCN* expression levels (Fig. 1i), as well as *ALYREF* copy number (Fig. 1j), further supporting ALYREF as a key intermediary in the cooperation between 17q21-ter gain and *MYCN*-amplification in the malignant neuroblastoma phenotype. By further examining pan-cancer cell line gene expression (RNA-seq) and copy number profiles using Cancer Cell Line Encyclopedia (CCLE)/Cancer Dependency Map[33], we observed that neuroblastoma cell lines had the highest *ALYREF* gene expression and copy number compared with all other cancer types, supporting the principle that *ALYREF* is an abundant gene target in neuroblastoma (Supplementary Fig. 1g, h).

**MYCN increases neuroblastoma cell viability in an ALYREF-dependent manner.** To investigate the functional relationship between MYCN and ALYREF in vitro, we evaluated the

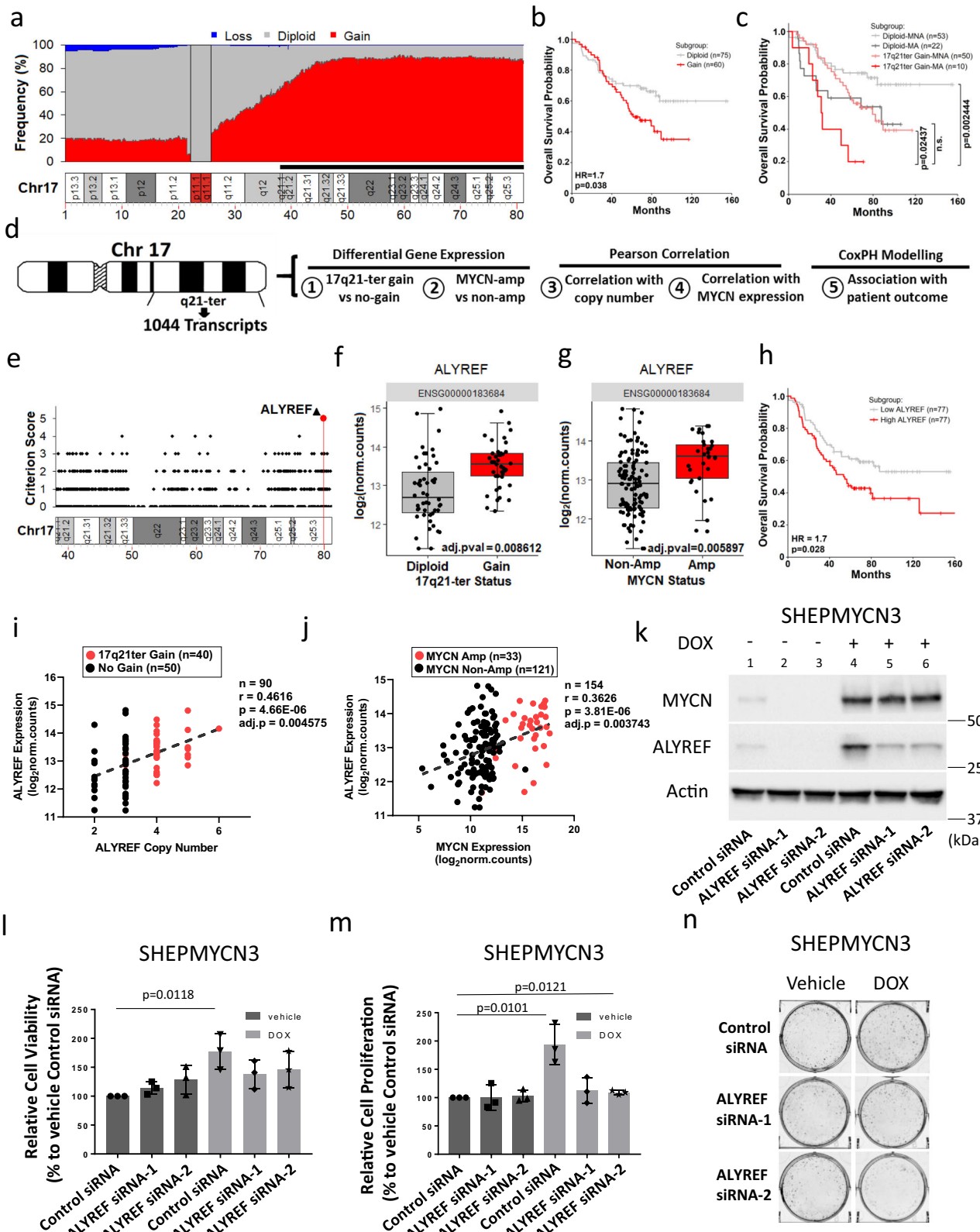

phenotypic effect of ALYREF knockdown on human SHEP-MYCN3 neuroblastoma cells, which permits doxycycline-inducible overexpression of stably incorporated MYCN, and are therefore ideal for MYCN gene dependency experiments. SHEPMYCN3 cells were transfected with control siRNA, ALYREF siRNA-1, or ALYREF siRNA-2 (Fig. 1k) for 72 h with (lanes 4–6) and without doxycycline (lanes 1–3). As expected,

transfection of ALYREF siRNA in MYCN-low and ALYREF-low expressing conditions did not reduce cell viability (Fig. 1l) or proliferation (Fig. 1m, n and Supplementary Fig. 1i). However, MYCN-induced increases in viability (Fig. 1l) and proliferation (Fig. 1m–n and Supplementary Fig. 1i) were blocked by ALYREF knockdown. Taken together these data showed that MYCN-induced increases in neuroblastoma cell viability and proliferation

**Fig. 1 High *ALYREF* expression associates with 17q21-ter gain, *MYCN*-amplification, and poor patient prognosis in neuroblastoma. a** Frequency of segmental copy number calls (loss, diploid, gain) spanning Chromosome 17 (10 kilobase genomic bins), from whole-genome sequencing (WGS) of the TARGET cohort ($n = 135$)[25]. 17q21-ter region is annotated by a black bar, genomic coordinates are provided in megabases. Centromeric regions were excluded (p/q11.1). Overall survival based on **b** 17q21-ter stratification and **c** 17q21-ter (gain or diploid) and *MYCN* status (Amplified; MA, Non-Amplified; MNA) in the TARGET cohort ($n = 135$). Hazard ratio (HR) from a Cox proportional hazards (CoxPH) model. *p*-values are from a two-sided log-rank test, and adjusted using the Benjamini–Hochberg method. **d** RNA-seq ($n = 154$) and WGS ($n = 135$) data of the TARGET cohort[25] were utilized to analyse1044 genes on the 17q21-ter region. **e** Scores of the 1044 genes according to genomic location, genomic distances in megabases (Mb). *ALYREF* expression between **f** diploid ($n = 50$) and 17q21-ter gain ($n = 40$) and **g** *MYCN* non-amplified ($n = 121$) and *MYCN*-amplified ($n = 33$) patients adjusted *p*-value is from two-sided Wald chi-squared tests, followed by Benjamini–Hochberg method in **f** and **g**. **h** Overall survival of the TARGET cohort ($n = 154$) dichotomized by median *ALYREF* expression (RNA-Seq). Hazard ratios (HR) and two-sided log-rank *p*-values are from a univariate CoxPH model. Correlation between **i** *MYCN* and *ALYREF* expression RNA-seq ($n = 154$), and **j** *ALYREF* expression RNA-seq and *ALYREF* copy number WGS ($n = 90$), *MYCN*-amplified patients are annotated in red. *p*-values from two-sided Pearson's correlation test in **i** and **j**. **k** Immunoblot for MYCN and ALYREF following ALYREF knockdown in doxycycline-induced (+) vs non-induced (−) SHEPMYCN3 cells. **l** Cell viability ($n = 3$), **m** proliferation ($n = 3$), and **n** colony formation analysis of DOX vs Vehicle-treated SHEPMYCN3 cells transfected with ALYREF siRNAs. *p*-values from two-sided Student's *t*-test in **l** and **m**. Data are representative of three independent experiments in **k** and **n**. Data are shown as mean ± s.e.m. (error bars) and representative of three independent experiments in **l** and **m**. The line represents the median expression value, and the upper/lower bounds represent the interquartile range of all expression values in **f** and **g**. The whiskers represent 1.5× the interquartile range of all expression values, from the upper/lower bounds in **f** and **g**. Panel **d** by J. Seneviratne.

were, in part, dependent on ALYREF. ALYREF is known to regulate cell proliferation[23,24], however, its role in regulating cell survival is unknown. To determine whether ALYREF knockdown affects MYCN-induced increases in cell survival, we investigated the effect of ALYREF inhibition on cell death in doxycycline-induced (DOX) or not induced (Vehicle) SHEPMYCN3 cells, transfected with control siRNA, ALYREF siRNA-1, or ALYREF siRNA-2 (Supplementary Fig. 1j–k). Abolition of mitochondrial membrane potential is a marker of apoptosis. Mitochondrial membrane potential was measured using a MitoProbe DilC1(5) Assay (Supplementary Fig. 1j). MYCN overexpression induced by doxycycline treatment did not affect mitochondrial membrane potential (Supplementary Fig. 1j) at the earlier time points. Only at 48 h was there a slight decrease in mitochondrial membrane potential induced by MYCN overexpression. ALYREF knockdown did not significantly affect the survival of the doxycycline-induced (DOX) or not induced (Vehicle) SHEPMYCN3 cells. To further investigate necrotic processes consequent on altered ALYREF levels, we assessed the effects of ALYREF knockdown on MYCN-induced survival by measuring SYTOX Green accumulation in the cells (Supplementary Fig. 1k). The dye is able to penetrate (and then bind to the nucleic acids) only necrotic cells with ruptured plasma membranes, whereas healthy cells with intact surface membranes show significantly lower SYTOX Green staining. As expected, doxycycline-induced MYCN overexpression protected the cells from cell death and significantly decreased the number of necrotic SHEPMYCN3 cells (Supplementary Fig. 1k). However, ALYREF knockdown did not further decrease SYTOX Green accumulation in the cells. Therefore, we conclude that ALYREF predominantly causes induction of proliferation in MYCN overexpressing cells, but does not assist MYCN in regulating cell survival. We also investigated the phenotypic effect of ALYREF knockdown on several human neuroblastoma cell lines with high ALYREF expression (Supplementary Fig. 1l–m). Endogenous knockdown of ALYREF in all neuroblastoma cells (Supplementary Fig. 1l) was accompanied by a significant decrease in cell viability (Supplementary Fig. 1m), as measured by Alamar Blue assay.

**MYCN directly upregulates ALYREF transcription.** We next investigated the regulatory relationship between MYCN and ALYREF. High *ALYREF* mRNA expression correlated with *MYCN* mRNA expression in the TARGET[25] ($n = 154$; Fig. 1j), SEQC[29–32] ($n = 498$; Fig. 2a), and Kocak[34] ($n = 476$; Supplementary Fig. 2a) neuroblastoma patient tumor data sets. ALYREF

protein expression levels were high in *MYCN*-amplified and MYCN overexpressing human neuroblastoma cells (lanes 1–7) across a panel of neuroblastoma cell lines (Fig. 2b–c), with significantly higher expression levels when compared to human fibroblasts (lanes 12 and 13; WI-38 and MRC5; Fig. 2b, c). Interestingly, two *MYCN*-non-amplified cell lines, with high cMYC expression (lanes 8 and 10; SH-SY5Y and SK-N-AS), also had significantly higher expression levels of ALYREF, when compared to human fibroblasts (lanes 12 and 13; WI-38 and MRC5; Fig. 2b–c). ALYREF knockdown in SH-SY5Y and SK-N-AS neuroblastoma cells (Supplementary Fig. 1l) was accompanied by a significant decrease in cell viability (Supplementary Fig. 1m), indicating a possible regulatory relationship between cMYC and ALYREF. Transfection of SH-SY5Y and SK-N-AS cells (Supplementary Fig. 2b–d) with cMYC siRNA-1 or cMYC siRNA-2 efficiently knocked down cMYC mRNA (Supplementary Fig. 2b) and protein (Supplementary Fig. 2c, d), and reduced *ALYREF* mRNA (Supplementary Fig. 2b) and protein expression (Supplementary Fig. 2c, d).

We next sought evidence that *ALYREF* mRNA expression in the organ of tumor origin, sympathetic ganglia, correlated with an *MYC* expression signature from tumor initiation to progression in tissues from the TH-*MYCN*[+/+] transgenic neuroblastoma mouse model[35]. Microarray mRNA expression analysis showed increasing *ALYREF* and *MYC*-signature gene expression throughout tumor progression in ganglia isolated from TH-*MYCN*[+/+] mice in comparison to ganglia from wild-type littermates (Fig. 2d). Importantly, high *ALYREF* mRNA expression strongly correlated with *MYCN* expression in ganglia tissues throughout tumor progression (Fig. 2e). Analysis of the *ALYREF* gene promoter sequence and published chromatin-immunoprecipitation (ChIP)-sequencing data[36] revealed non-canonical MYCN binding sites CACCTG (at −549 bp) and CAGCTG (+635 bp) in close proximity to the *ALYREF* transcription start site (TSS, Fig. 2f). We performed ChIP assays with an anti-MYCN antibody or control mouse IgG antibody (Santa Cruz Biotechnologies) followed by real-time PCR with primers targeting a negative control (Control; 2000 bp upstream TSS) and the MYCN peak summit (Promoter, −248 bp) upstream of the *ALYREF* TSS (Fig. 2g). ChIP assays showed significant MYCN binding[14,37,38] at its putative binding site 248 bp upstream *ALYREF* TSS. The MYCN antibody immunoprecipitated the *ALYREF* region 3-fold higher than the negative control region in SK-N-BE(2)C cells and 6-fold in Kelly cells (Fig. 2h), confirming that MYCN directly binds the *ALYREF* gene promoter. Transfection of *MYCN*-amplified SK-N-BE(2)C, Kelly (Fig. 2i–j) and CHP-134 cells (Supplementary

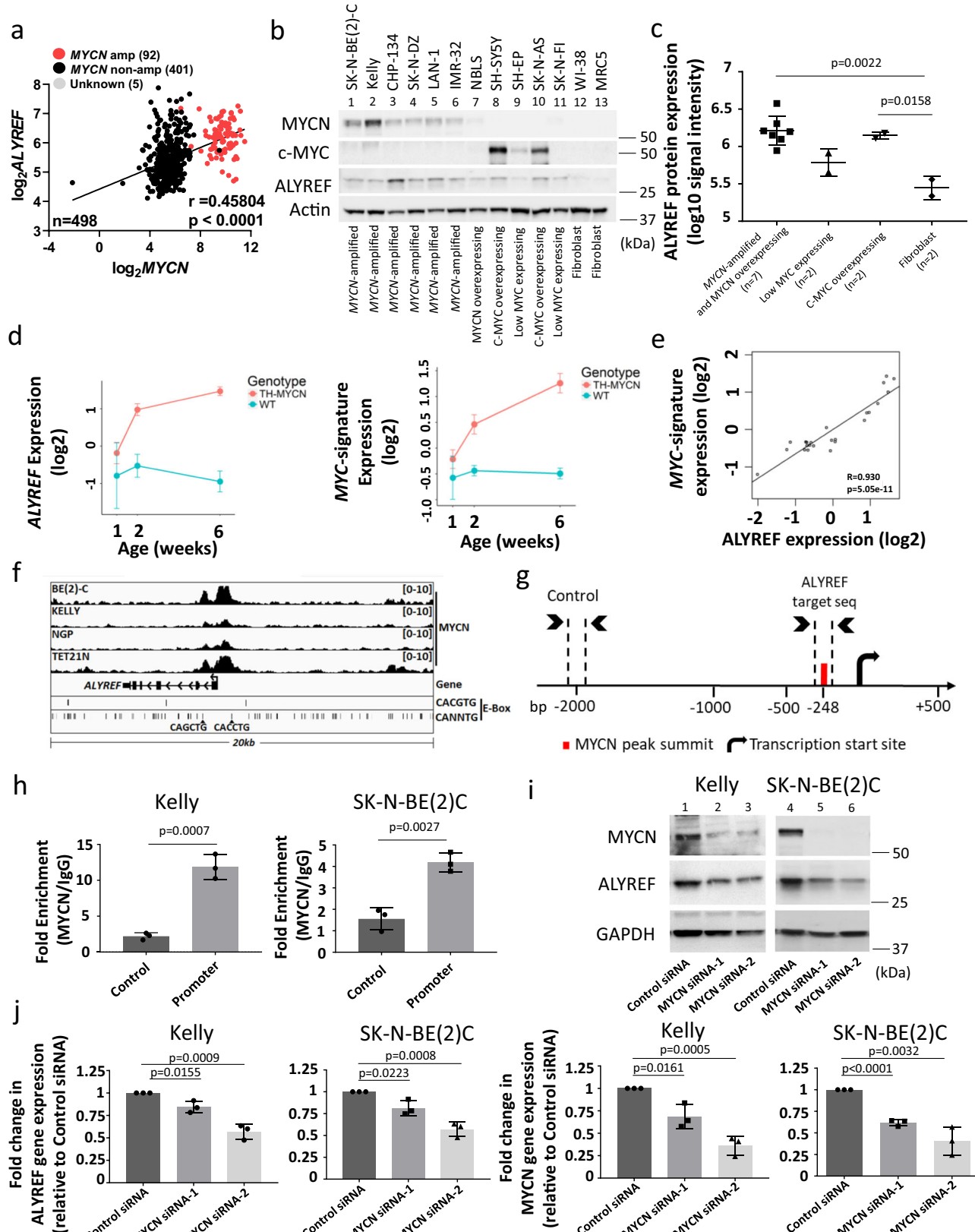

Fig. 2e, f) with MYCN siRNA-1 or MYCN siRNA-2 (lanes 2–3 and lanes 5–6) efficiently knocked down MYCN protein (Fig. 2i and Supplementary Fig. 2e) and mRNA (Fig. 2j and Supplementary Fig. 2f), and, reduced ALYREF mRNA and protein expression (Fig. 2i, j and Supplementary Fig. 2e, f). Thus, *ALYREF* is a transcriptional target of MYCN in *MYCN*-amplified neuroblastoma cells.

**ALYREF participates in a positive interconnected regulatory signaling axis with MYCN.** We next assessed whether ALYREF acted in a forward feedback expression loop with MYCN, as we have identified for other MYCN transactivation targets[37–40]. Silencing endogenous ALYREF with ALYREF siRNA-1 or ALYREF siRNA-2 (lanes 2–3 and 5–6, Fig. 3a) in

**Fig. 2 MYCN directly upregulates *ALYREF* transcription. a** Scatter plot, with a linear regression fit for 498 neuroblastoma patients from the SEQC cohort for *MYCN* vs *ALYREF* gene expression (RNA-Seq, $\log_2$RPM). Patients with *MYCN*- amplification, no amplification, or are of unknown status are annotated. A two-sided Pearson's correlation test was used to derive *p*-values. **b** Immunoblot and **c** densitometry analysis (*n* = 3 per group) for ALYREF expression in a range of *MYCN*-amplified (SK-N-BE(2)C, Kelly, CHP-134, SK-N-DZ, LAN-1, IMR-32) and MYCN overexpressing (NBLS), low MYC or MYCN expressing (SH-EP, SK-N-FI), cMYC overexpressing neuroblastoma (SH-SY5Y, SK-N-AS) and normal lung fibroblast (WI-38, MRC5) cells. Differences in ALYREF protein expression were compared using two-sided one-way ANOVA. **d** mRNA expression of *MYC*-signature[35] genes and *ALYREF* in ganglia from *TH-MYCN*[+/+] mice, obtained at different postnatal ages (1, 2, and 6 weeks of age; *n* = 4 per group; microarray, $\log_2$). **e** Scatter plot, with a linear regression fit from the *TH-MYCN*[+/+] cohort for *MYC*-signature vs *ALYREF* gene expression (6 weeks of age; microarray, $\log_2$). A two-sided Pearson's correlation test was used to derive *p*-values. **f** Schematic of the *ALYREF* promoter in *MYCN*-amplified neuroblastoma cells detailing canonical (CACGTG) and non-canonical (CANNTG) MYCN binding motifs. **g** Schematic of the *ALYREF* (Promoter) and negative control (Control) target sequences used for ChIP-PCR, detailing the MYCN peak summit and its distance from the transcription start site (TSS). **h** ChIP-PCRassays (*n* = 3 per group) for the negative control region (Control; 2000 bp downstream of TSS) or the *ALYREF* promoter containing the MYCN peak summit (Promoter, −248 bp downstream of TSS). Two-sided unpaired Student's *t*-tests were performed to derive *p*-values. Differences in expression levels were compared to the negative control (Control). **i** Immunoblots **j** and RT-PCR (*n* = 3 per group) of MYCN and ALYREF expression following MYCN knockdown. Two-sided unpaired Student's *t*-tests were performed to derive *p*-values. Differences in expression levels were compared to Control siRNA. Pearson correlation coefficient (*r*) and respective *p*-values (*p*) are displayed. Comparisons were not significant unless otherwise noted. Data is representative of three independent experiments with similar results in **b** and **i**. Data are shown as mean ± s.e.m. (error bars) and representative of three independent experiments in **c**, **d**, **h**, and **j** and 4 mice per age group in **d**.

*MYCN*-amplified neuroblastoma cells significantly decreased MYCN protein expression (Fig. 3a), however, ALYREF knockdown had no effect on *MYCN* mRNA levels (Fig. 3b). Stable inhibition of ALYREF by ALYREF shRNA (lanes 6–10, Fig. 3c) in cycloheximide-treated, *MYCN*-amplified neuroblastoma cells reduced MYCN protein half-life from 36–40 to 14–15 min (Fig. 3c, d). Doxycycline treatment had minimal effect on MYCN half-life in doxycycline-inducible control shRNA Kelly and SK-N-BE(2)C cells (Supplementary Fig. 3a). Similarly, transient over-expression of ALYREF (Supplementary Fig. 3b) significantly increased the half-life of MYCN protein in *MYCN*-amplified SK-N-BE(2)C cells (Supplementary Fig. 3c). These data suggest that ALYREF-regulated MYCN expression through a post-translational mechanism. MYCN goes through rapid degradation by the ubiquitin-proteasome system[41]. To examine the hypothesis that ALYREF-regulated MYCN half-life through ubiquitination in neuroblastoma cells, we transfected Kelly and SK-N-BE(2)C cells with control siRNA, ALYREF siRNA-1, or ALYREF siRNA-2. Following treatment with the proteasome inhibitor, MG132, the total cellular protein was immunoprecipitated for endogenous MYCN[42] and then probed with an antibody specific for K-48-linked ubiquitin chains, the trigger signal for proteasomal degradation (Fig. 3e). The results showed that ALYREF knockdown increased K-48-linked ubiquitination of MYCN (lanes 5–6 vs lane 4; Fig. 3e) in both cell lines. These data indicate that ALYREF maintains a high level of MYCN expression through inhibition of its protein degradation.

**ALYREF forms a protein complex with MYCN in the nucleus.** Since ALYREF modulated MYCN protein stability, we examined the possibility of their physical interaction. We transfected HEK293T human embryonic kidney cells with a construct expressing HA-tagged MYCN either alone, or one expressing Flag-tagged ALYREF. An MYCN co-immunoprecipitation (co-IP) of total cellular protein with an HA-tag antibody showed that ALYREF-bound MYCN (lane 6; Fig. 4a), which was confirmed in reciprocal ALYREF co-IP experiments with a Flag-tag antibody (lane 6; Fig. 4b). Separation of total cellular protein into nuclear and cytoplasmic fractions by centrifugation allowed us to show that both MYCN and ALYREF were predominantly localized to the nucleus (lanes 3 and 6; Fig. 4c) in *MYCN*-amplified neuroblastoma cell lines (Kelly and SK-N-BE(2)C). We next examined total cellular protein (lanes 1–3 and 10–12; Fig. 4d, Supplementary Fig. 4a) as well as nuclear (lanes 7–9 and 16–18; Fig. 4d) and cytoplasmic extracts (lanes 4–6 and 13–15; Fig. 4d) which were immunoprecipitated with an MYCN-specific antibody and found

that endogenous nuclear ALYREF bound to the immunoprecipitated MYCN (lanes 9 and 18; Fig. 4d) in Kelly and SK-N-BE(2)C cells. To further validate this interaction, we performed the reciprocal binding assay and showed that immunoprecipitated endogenous ALYREF-bound endogenous MYCN (lanes 3 and 6; Supplementary Fig. 4b).

To map the domain of MYCN, which was critical for the interaction with ALYREF, we used MYCN deletion mutant expression constructs[43] (Fig. 4e). We co-transfected HEK293T-cells with constructs expressing HA-tagged MYCN deletion mutants and a construct expressing Flag-tagged full-length ALYREF. Flag immunoprecipitation assays showed that ALYREF was present in the protein immunoprecipitates containing HA-tagged full-length MYCN (amino acids 1–464), MYCN (Δ1–123), MYCN (Δ382–464), MYCN (Δ346–464), and MYCN (281–464) (lanes 8–11; Fig. 4f). In contrast, ALYREF was not detected in the immunoprecipitates containing HA-MYCN (Δ281–464) by the same approach (lane 12; Fig. 4f), even following prolonged exposure of immunoblots (Supplementary Fig. 4c). These data demonstrate that a small region of MYCN containing amino acids 281–345 was crucial for the binding to ALYREF.

**ALYREF and MYCN form a transcriptional activator complex which upregulates *USP3* expression.** ALYREF has been suggested to act as a regulator of DNA binding and modulate the activity of transcription factors[15–18]. The binding of ALYREF and MYCN proteins in the nucleus, as well as the positive forward feedback expression loop between the two proteins, led us to hypothesize that ALYREF and MYCN cooperated and played a role in upregulating transcriptional programs involved in the ubiquitination process regulating MYCN turnover. To identify gene targets, we first mapped the genome-wide profile of ALYREF-chromatin interactions in SK-N-BE2C cells using ChIP-seq. We looked for transcriptional target genes of ALYREF based on the following selection criteria (Fig. 5a): (1) presence of MYCN and RNA Pol II (indicating active transcriptional elongation), as well as two proximate histone modifications associated with transcriptional activation (H3K27ac and H3K4me3) at the gene promoter; (2) having a high fold enrichment (FE) for ALYREF binding of more than 5; (3) expression correlation with neuroblastoma patient survival in the SEQC neuroblastoma patient tumor cohort (*n* = 498); (4) correlation with *MYCN* expression level in the SEQC neuroblastoma patient tumor cohort (*n* = 498); and (5) having known biological function relevant to ubiquitination. We found a total of 3598 ALYREF-binding sites (FDR < 0.05; Supplementary Fig. 5a). We identified 194

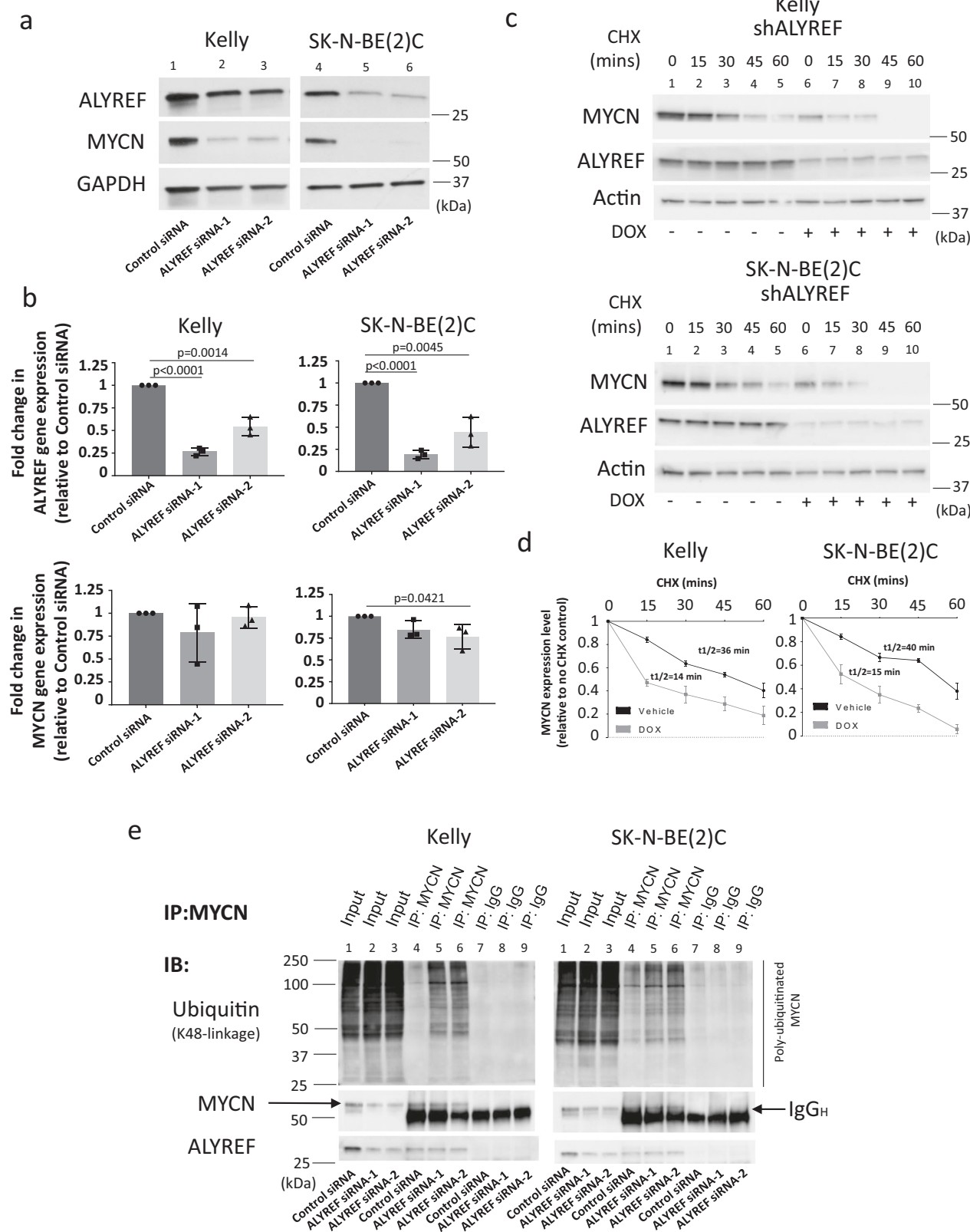

ALYREF-regulated genes undergoing transcription, among which 57 genes (Supplementary Table 4) had a FE of more than 5 (Fig. 5b). From these 57 genes, we identified the ubiquitin-specific protease 3 gene (*USP3*) as satisfying all criteria. *USP3* had FE = 6.15987 for ALYREF binding and was ranked as number 6 among the 57 genes (Supplementary Table 4) based on fold enrichment

stratification. Integrative Genomic Viewer showed that ALYREF bound the *USP3* intronic region 6.8 kb from the TSS (Fig. 5c). To confirm the ChIP-seq data, we performed a ChIP assay with an anti-ALYREF antibody or control IgG antibody (IgG) followed by real-time PCR with primers targeting a negative control region (Control; 2000 bp downstream TSS) or the putative

**Fig. 3 ALYREF regulates MYCN stability in a forward feedback expression loop. a** Immunoblot and **b** real-time PCR analysis ($n = 3$ per group) of MYCN and ALYREF expression in Kelly and SK-N-BE(2)C cells following siRNA-mediated ALYREF knockdown for 24 h. Two-sided unpaired Student's $t$-tests were performed to derive $p$-values. Differences in expression levels were compared to the control siRNA. **c** Kelly and SK-N-BE(2)C cells expressing ALYREF shRNA (shALYREF) were treated with doxycycline (DOX; 2 μg/ml) for 72 h, followed by treatment with 100 ug/ml cycloheximide (CHX) for 0 (no CHX), 15, 30, 45, or 60 min. Protein was extracted from the cells and subjected to immunoblot analysis of MYCN and ALYREF expression. **d** Densitometry analysis ($n = 3$ per group) of CHX assay. MYCN protein levels were normalized by actin, the ratio of MYCN protein/actin protein was artificially set as 1.0 for samples untreated with CHX to obtain half-life ($T_{1/2}$) of MYCN. $p$-values for 60 min time point ($p = 0.024$ for Kelly and $p = 0.0039$ for SK-N-BE(2)C) derived from two-sided $t$-test. **e** Kelly and SK-N-BE(2)C cells were transfected with ALYREF-specific siRNAs (ALYREF siRNA-1 or ALYREF siRNA-2) as well as control siRNA for 48 h, followed by treatment with MG132 for 4 h, then subjected to endogenous MYCN immunoprecipitation and immunoblot analyses of K-48-mediated ubiquitination. Respective $p$-values ($p$) are displayed. Comparisons were not significant unless otherwise noted. Data are representative of three independent experiments with similar results in **a**, **c**, and **e**. Data are shown as mean ± s.e.m. (error bars) and representative of three independent experiments in **b** and **d**.

ALYREF-bound region located downstream of the *USP3* TSS (Fig. 5d). The ChIP assay showed that the ALYREF antibody immunoprecipitated the ALYREF-bound region in both Kelly and SK-N-BE(2)C cells (Supplementary Fig. 5b). We also performed single ChIP and a sequential double ChIP assay[44,45] with anti-MYCN (MYCN; single ChIP) or anti-MYCN and anti-ALYREF antibodies (MYCN:ALYREF; double ChIP), followed by real-time PCR with primers targeting either a negative control region (Control, −2000 bp from TSS) or the ALYREF-bound region located 6.8 kb downstream of the *USP3* TSS (Fig. 5d). The antibody against MYCN efficiently immunoprecipitated the intronic region of *USP3* carrying the ALYREF-binding sites at 6.8 kb from the TSS and upstream from *USP3* exon 1 (Fig. 5e). Finally, double ChIP for an MYCN-ALYREF complex immunoprecipitated the ALYREF-bound intronic site of the *USP3* gene some 3-fold higher than the negative control region in both Kelly and SK-N-BE(2)C cells, respectively (Fig. 5e). The binding of ALYREF to MYCN on chromatin might be due to a yet unknown function of ALYREF in regulating long-range interactions involving chromatin looping. To confirm the potential interaction between the *USP3* promoter sequence binding MYCN at the TSS and the DNA sequence containing the ALYREF-binding site at +6800 bp from the TSS, a chromatin conformation capture (3C) assay (Fig. 5f) was performed using restriction enzyme BamH1 (Supplementary Fig. 5c). A specific PCR product (PCR amplicon#1) was detected in BamH1 digested SK-N-BE(2)C samples (Fig. 5g and Supplementary Fig. 5d). Genomic DNA obtained from SK-N-BE(2)C cells without cross-linking (Fig. 5f) was used as a positive control (PCR amplicon#2) in these experiments (Fig. 5f–g and Supplementary Fig. 5d). The DNA looping detected by the 3C assay was further validated using an additional restriction enzyme HindIII, with a restriction site in the ALYREF peak (Supplementary Fig. 5c). Restriction digestion with *Hin*dIII will prevent the ligation of the MYCN binding and ALYREF-binding sequences, resulting in no PCR product using PCR amplicon #1 primers. As expected, when both *Hin*dIII and *Bam*H1 were used, using the same PCR amplicon #1 primers pairs, no PCR product was obtained (Fig. 5g and Supplementary Fig. 5d). To confirm the 3C data and validate the assay, the PCR products were run on a DNA High Sensitivity (HS) chip using the Agilent 2100 Bioanalyzer (Supplementary Fig 5e). The amplified product was not detected from the sample in which both restriction enzymes used but was present with BamH1 alone (Supplementary Fig 5e). These results suggest that the two DNA fragments are potentially interacting or are in close proximity with each other. To further examine ALYREF as a transcriptional coactivator for MYCN, we evaluated the effect of ALYREF knockdown on *USP3* gene transcription in SHEPMYCN3 neuroblastoma cells, which express MYCN under the control of doxycycline, therefore minimising ALYREF knockdown effects on MYCN stability. SHEPMYCN3 neuroblastoma cells were transfected with control siRNA, ALYREF

siRNA-1, or ALYREF siRNA-2 for 72 h, with and without doxycycline. Transfection of ALYREF siRNA in MYCN-low and ALYREF-low expressing conditions did not affect USP3 mRNA (Fig. 5h and Supplementary Fig. 5f) or protein (lanes 2–3; Fig. 5i) expression, however, the MYCN-induced increase in USP3 mRNA (Fig. 5h and Supplementary Fig. 5f) or protein (lanes 5–6; Fig. 5i) levels were diminished by ALYREF knockdown. These data showed that the MYCN-driven transactivation of *USP3* was dependent on ALYREF expression. Importantly, we were able to replicate these findings in *MYCN*-amplified Kelly and SK-N-BE(2)C cells. Transfection with ALYREF siRNA-1 or ALYREF siRNA-2 in SK-N-BE(2)C cells, and ALYREF siRNA-2 in Kelly cells, efficiently knocked down *USP3* mRNA (Fig. 5j). Together these data suggest MYCN and ALYREF bind together in a transcriptional coactivator complex to increase *USP3* transcript levels, consequently reducing MYCN ubiquitination, and further increasing MYCN protein to the levels required to drive neuroblastoma tumorigenesis.

**USP3 regulates MYCN expression through its deubiquitinase activity.** Consistent with an MYCN-ALYREF-USP3 signal driving neuroblastoma tumorigenesis, we found that high *USP3* mRNA expression in the SEQC neuroblastoma patient cohort ($n = 498$) also correlated with poor prognosis (Fig. 6a and Supplementary Fig. 6a). Importantly, high *USP3* mRNA expression correlated with *MYCN* expression (Fig. 6b). Multivariate analysis for both overall (Fig. 6c) and event-free (Supplementary Fig. 6b) survival showed that high *USP3* expression had no independent prognostic significance when compared with other prognostic factors such as patient age, clinical stage, and *MYCN*-amplification. Further analysis for both overall (Fig. 6c) and event-free (Supplementary Fig. 6c) survival showed that *MYCN*-amplification and *ALYREF* expression had the greatest negative effect on *USP3*'s prognostic significance, suggesting that the role of *USP3* in the malignant neuroblastoma cell phenotype is dependent on the MYCN-ALYREF regulatory complex. By further examining pan-cancer cell line gene expression (RNA-seq) profiles using Cancer Cell Line Encyclopedia (CCLE)/Cancer Dependency Map[33] we observed that neuroblastoma cell lines had the highest *USP3* gene expression compared with all other cancer types (Supplementary Fig. 6d). We saw significant positive correlations between *USP3* and *MYCN* expression levels (Supplementary Fig. 6e) in neuroblastoma tissues. We also observed that neuroblastoma cell lines showed marked dependency on *USP3* for cell viability (Fig. 6d), especially in the *MYCN*-amplified context (Fig. 6e). High endogenous USP3 protein expression was detected in total cellular protein from six *MYCN*-amplified (lanes 1–6), one MYCN overexpressing (lane 9), and two cMYC expressing cell lines (lanes 7 and 8; Fig. 6f), but not in three neuroblastoma cell lines without MYCN or cMYC expression or normal fibroblasts (lanes 10–13; Fig. 6f). USP3 has been shown to target

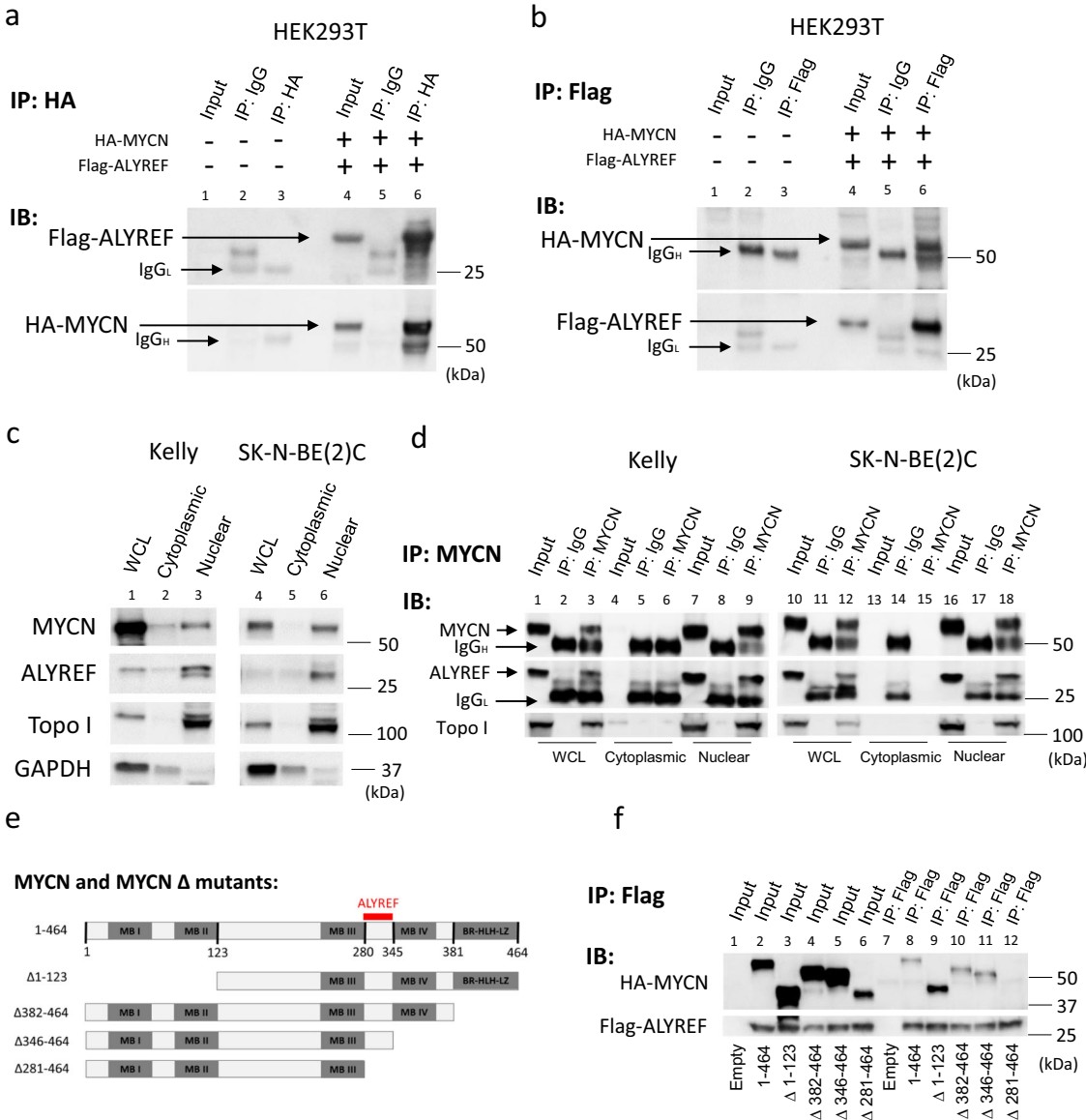

**Fig. 4 ALYREF forms a protein complex with MYCN in the nucleus.** Representative western blot analysis for the indicated ectopically overexpressed HA-MYCN and Flag-ALYREF from HEK293T cells after immunoprecipitation of **a** HA-MYCN or **b** Flag-ALYREF. 5% of the cell lysate was loaded for input. "-" indicates HEK293T cells that were transfected with empty vectors. **c** Representative Western blot analysis for endogenous ALYREF and MYCN localization in Kelly and SK-N-BE(2)C cells following nuclear and cytoplasmic fractionation. WCL whole-cell lysate; Topo I, Topoisomerase I. Topoisomerase I, and GAPDH were used as nuclear and cytoplasmic fraction markers. **d** Representative Western blot analysis for endogenous ALYREF after immunoprecipitation of endogenous MYCN from Kelly and SK-N-BE(2)C cells following nuclear and cytoplasmic fractionation. 5% of the cell lysate was loaded for input. WCL, whole-cell lysate. Topo I, Topoisomerase I. Topoisomerase I was used as a nuclear marker. **e** Schematic representation of the HA-MYCN deletion mutants. MB I, MYC box I; MB II, MYC box II; MB III, MYC box III; MB IV, MYC box IV; BR, basic region; HLH, helix-loop-helix; LZ, leucine zipper. **f** Representative Western blot analysis for the indicated ectopically overexpressed HA-MYCN deletion mutants and Flag-ALYREF from HEK293T cells after immunoprecipitation of Flag-ALYREF. 5% of the cell lysate was loaded for input. Empty indicates HEK293T cells that were transfected with an empty vector. Data are representative of three independent experiments with similar results in **a**–**f**.

K-63-linked ubiquitin chains and regulate p53[46], CHK1[47], KLF5[48], and SUZ12[49] stability. USP3 interacts directly with its protein targets, which prompted us to examine whether MYCN also bound USP3 directly. Total cellular protein extracts from the *MYCN*-amplified neuroblastoma cell lines (Kelly and SK-N-BE(2) C) were immunoprecipitated with an MYCN-specific antibody. Endogenous USP3 was detected in the MYCN immunoprecipitates (lanes 3 and 6; Fig. 6g), but not with those using the control IgG antibody (lanes 2 and 4; Fig. 6g). To understand the functional consequence of this interaction, we measured the effects of USP3 expression on MYCN protein levels. We found that MYCN

protein levels increased after USP3 overexpression (Supplementary Fig. 6f) in a dose-dependent manner. Furthermore, an ectopically expressed USP3 mutant (Fig. 6h) lacking the catalytic activity (USP3$^{C168S}$)[46,47,49,50] had a significantly reduced effect on MYCN protein expression (Fig. 6i), suggesting that USP3 regulates MYCN expression through its deubiquitinase activity. Although both ectopically expressed USP3$^{wt}$ and USP3$^{C168S}$ coimmunoprecipitated with MYCN (Fig. 6j), the half-life of MYCN was only significantly extended in the presence of USP3$^{wt}$ (lanes 5–8) but not USP3$^{C168S}$ (lanes 9–12) or an empty (lanes 1–4) expression vector (Fig. 6k and Supplementary Fig. 6g).

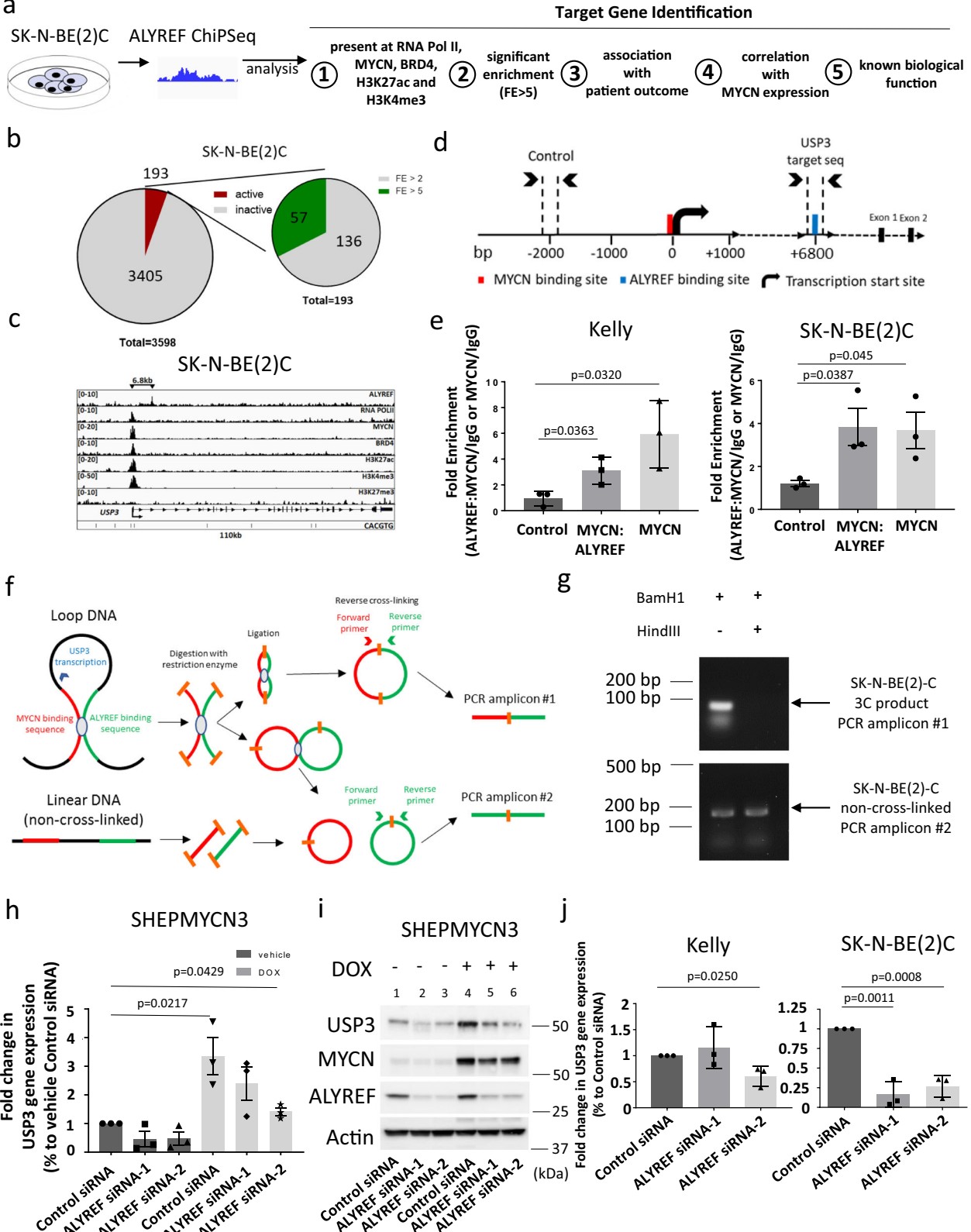

Despite their interaction, the USP3$^{C168S}$ mutant was unable to catalyze the de-ubiquitination of MYCN (lane 5) in HEK293T cells (Fig. 6l), suggesting that the catalytic activity is required for USP3 to impact MYCN ubiquitination levels. Collectively, these results demonstrate that MYCN is a target of USP3.

**USP3 is responsible for ALYREF-induced MYCN stability.** We further investigated the phenotypic effect of USP3 knockdown in *MYCN*-amplified neuroblastoma cell lines with high USP3 expression. Silencing endogenous USP3 with USP3 siRNA-1 or USP3 siRNA-2 (lanes 2–3 and 5–6, Fig. 7a) in *MYCN*-amplified neuroblastoma cells significantly decreased MYCN protein

**Fig. 5 ALYREF and MYCN form a transcriptional activator complex which upregulates *USP3* expression. a** Approach to identifying ALYREF target gene candidates in the context of MYCN stability regulation. ALYREF ChIP-seq data was used to analyze 3598 ALYREF-bound chromatin regions. **b** Genome-wide profile of ALYREF-chromatin interactions. **c** Integrative Genomic Viewer (IGV) representation of the *USP3* gene promoter detailing MYCN and ALYREF-binding motifs. **d** Schematic representation of the *USP3* and negative control target sequences used for ChIP-PCR, detailing MYCN and ALYREF-binding motifs and their distance from the transcription start site (TSS). **e** Single ChIP assays for MYCN or double ChIP assays for MYCN:ALYREF complex analyzed by RT-PCR ($n = 3$ per group) with primers targeting the negative control (Control) region or the *USP3* gene. Two-sided unpaired Student's $t$-tests were performed to derive p-values. Differences in expression levels were compared to the negative control (Control). **f** Schematic of chromatin conformation capture (3C) assay used to detect the interactions between the DNA fragments containing the MYCN binding site at the *USP3* promoter region (red, 0 bp) and that containing the ALYREF peak (green, +6800 bp from *USP3* TSS). **g** Representative agarose gel from 3C assay performed with *Bam*H1 and/or *Hind*III showing PCR product of MYCN and ALYREF interaction (PCR Amplicon #1; 166 bp) and the positive control non-cross-linked DNA (PCR Amplicon #2; 53 bp). **h** RT-PCR ($n = 3$ per group) and **i** immunoblot analyses of USP3 (**h**) and USP3, ALYREF, and MYCN (**i**) expressions following siRNA-mediated ALYREF knockdown in SHEPMYCN3 cells. Two-sided unpaired Student's $t$-tests were performed to derive p-values. Differences in expression levels were compared to the vehicle-treated control siRNA. **j** RT-PCR analysis ($n = 3$ per group) of *USP3* expression following siRNA-mediated ALYREF knockdown. Two-sided unpaired Student's t-tests were performed to derive p-values. Differences in expression levels were compared to the control siRNA. Respective p-values ($p$) are displayed. Comparisons were not significant unless otherwise noted. Data are shown as mean ± s.e.m. (error bars) and representative of three independent experiments in **e**, **h**, and **j**. Data are representative of three independent experiments with similar results in **g** and **i**. Panels **a** and **f** created by J. Seneviratne and Z. Nagy.

expression (Fig. 7a). Knockdown of USP3 had marked effects on the cancer phenotype in *MYCN*-amplified neuroblastoma cells. Alamar blue assays showed that Kelly and SK-N-BE(2)C cells transfected with USP3 siRNA-1 or USP3 siRNA-2 displayed reduced cell viability (Fig. 7b) when compared to Control siRNA expressing cells. In addition, knocking down USP3 decreased both short-term (Fig. 7c) and long-term (Fig. 7d and Supplementary Fig. 7a) cell proliferation of Kelly and SK-N-BE(2)C cells. To investigate whether USP3 is responsible for the ALYREF-induced increase in MYCN stability and neuroblastoma cell growth, we created a pool of clones of Kelly and SK-N-BE(2)C cells (Fig. 7e) stably overexpressing either Vector control (Vector; lane 2) or USP3 (USP3; lane 3), and evaluated the effect of USP3 overexpression on MYCN ubiquitination and neuro-blastoma cell viability following ALYREF knockdown. Kelly and SK-N-BE(2)C cells stably overexpressing vector control (Vector) or USP3 were transfected with control siRNA, ALYREF siRNA-1, or ALYREF siRNA-2 (Fig. 7f, g). Alamar blue assays for cell viability showed that knocking down ALYREF reduced the number of viable Kelly and SK-N-BE(2)C cells, however, stable overexpression of USP3 reversed the effect of ALYREF knockdown (Fig. 7f). We next examined the hypothesis that the ALYREF effects on MYCN ubiquitination were USP3 dependent. As expected, knocking down ALYREF (lanes 2 and 3; Fig. 7g "INPUT") decreased USP3 and MYCN expression levels in both Kelly and SK-N-BE(2)C cells (lanes 2 and 3; Fig. 7g "INPUT"). USP3 overexpression, however, significantly increased MYCN expression levels in both cell lines (lanes 5 and 6; Fig. 7g "INPUT"). We immunoprecipitated endogenous MYCN using an MYCN antibody (lanes 1–6; Fig. 7g "IP") or control IgG antibody (lanes 7–12; Supplementary Fig. 7b "IP") from vector control (Vector) or USP3-overexpressing cells that had been treated with siRNAs for ALYREF then probed with antibodies specific for K-48- or K-63-linked ubiquitin chains. The results showed that ALYREF knockdown increased K-48- and K-63-linked poly-ubiquitination of MYCN (lanes 1–3; Fig. 7g "IP"), whereas forced overexpression of USP3 partially reversed this change in ubi-quitination (lanes 4–6; Fig. 7g "IP"). These findings suggest that USP3 is the effector protein for the effects of the MYCN-ALYREF-USP3 signal on neuroblastoma cell viability and MYCN stability.

**ALYREF is required for the growth and tumorigenicity of *MYCN*-amplified neuroblastoma cells.** We further explored the role of ALYREF in *MYCN*-amplified neuroblastoma cells by creating a pool of clones for doxycycline-inducible control

shRNA (shControl) or ALYREF shRNA (shALYREF) Kelly and SK-N-BE(2)C cells. Suppression of ALYREF resulted in a decrease in both ALYREF and MYCN expression levels (2000 ng/ ml doxycycline; lanes 4 and 8; Fig. 8a). Knockdown of ALYREF had profound effects on the cancer phenotype in neuroblastoma cells. Alamar blue assays showed that doxycycline-treated shA-LYREF Kelly and SK-N-BE(2)C cells displayed markedly reduced viability (Fig. 8b) even at lower doxycycline doses (500 ng/ml for SK-N-BE(2)C and 250 ng/ml for Kelly; Supplementary Fig. 8a). In addition, knocking down ALYREF decreased the percentage of Kelly and SK-N-BE(2)C cells in the S phase (Supplementary Fig. 8b) and their colony-forming capacity in vitro (Fig. 8c and Supplementary Fig. 8c). Importantly, these findings were repli-cated in vivo. SK-N-BE(2)C cells expressing ALYREF shRNA were xenografted into nude mice. Once tumors reached 5 mm in diameter, the mice were divided into doxycycline and vehicle control (Vehicle) subgroups, and fed with 10% sucrose water with or without 2 mg/ml doxycycline[35], until the tumor reached 1000 mm[3]. Subcutaneous tumor xenografts of SK-N-BE(2)C cells, transduced with doxycycline-inducible ALYREF shRNA, showed significant delays in tumor growth compared to vehicle controls (Fig. 8d). Kaplan–Meier survival analysis showed that doxycy-cline treatment resulted in longer median survival for mice xenografted with doxycycline-inducible ALYREF shRNA SK-N-BE(2)C cells and treated with doxycycline (Fig. 8e–f). Immuno-blot analysis of snap-frozen tumor tissues showed a decrease in ALYREF, USP3, and MYCN protein expression as a result of doxycycline treatment, compared to vehicle control (Fig. 8f–g and Supplementary Fig. 8d). These findings demonstrate that ALYREF promotes MYCN-directed tumorigenesis in vitro and in vivo.

## Discussion
Transcriptional regulators undergo rapid and timely degradation by ubiquitination in most normal eukaryotic cells. In cancer cells, MYC and MYCN exploit the ubiquitination pathway to stabilize aberrant oncogenic signaling and drive a wide variety of cellular processes required for carcinogenesis. Here, we have described a mechanism for MYCN degradation in MYCN-dependent neu-roblastoma, regulated by the ALYREF-USP3 signal.

ALYREF activates gene transcription by enhancing DNA-binding and activity[16–18] of transcription factors on target gene promoters. Our data suggest that the interaction between ALYREF and MYCN is likely to be important for efficient gene transcription. We confirmed that ALYREF protein directly binds to MYCN protein at the small region of amino acids 281–345

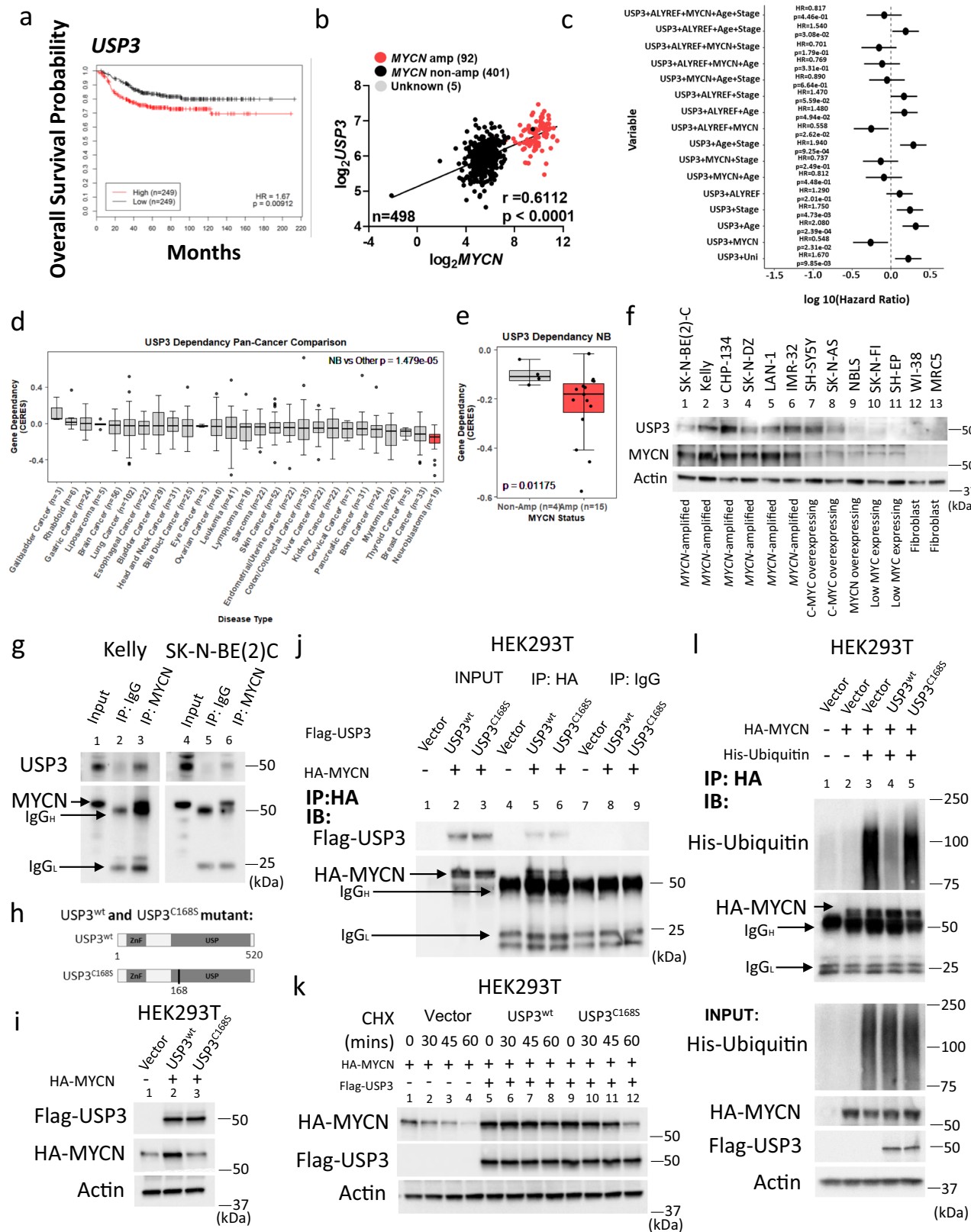

(AA 281–345) and the two proteins form a trans-activator complex to regulate *USP3* expression. The function of this domain in MYCN and MYC is still understudied. MYC proteins contain two highly conserved MYC boxes (MYC box III and IV)[51–53] in the central portion, with suggested roles in regulating DNA binding[54]. The ALYREF-binding site on MYCN lies just upstream of the BR/HLH/LZ domain in the primary amino acid sequence, but as the structure has not been fully solved for MYC proteins, the function and position of MYCN AA 281–345 in the secondary or tertiary structure are unknown. As MYCN showed dependency on ALYREF for efficient induction of *USP3* expression and the two proteins interact via DNA looping, we argue that

**Fig. 6** *USP3* expression associates with *MYCN*-amplification and predicts poor patient survival. **a** Overall survival of patients in the SEQC cohort ($n = 498$) when dichotomized by median *USP3* expression (RNA-Seq). Hazard ratios (HR) and two-sided log-rank p-values are presented from a univariate CoxPH model. **b** Scatter plot, with a linear regression fit for the SEQC cohort ($n = 498$) for *MYCN* vs *USP3* expression (RNA-Seq, log$_2$RPM). A two-sided Pearson's correlation test was used to derive *p*-values. **c** Multivariate overall survival ($n = 498$) dichotomized for advanced stage of disease, age of patient, *MYCN* amplification, *ALYREF* and *USP3* expression using cox regression modeling, *p*-values from two-sided log-rank tests. **d** *USP3* CRISPRi dependency expressed as CERES scores across 27 cancer types, *p*-value is from a two-sample *t*-test between neuroblastoma cell lines ($n = 718$) and an aggregate of all other cell lines. **e** *USP3* CERES scores for neuroblastoma (NB) divided into *MYCN*-non-amplified ($n = 4$) and *MYCN*-amplified ($n = 15$) cell lines, *p*-value is from a two-sample *t*-test. **f** Immunoblot for MYCN and USP3 in a range of neuroblastoma cells. **g** Immunoblot for USP3 after immunoprecipitation of MYCN. **h** Schematic of wild-type (USP3$^{wt}$) and mutant (USP3$^{C168S}$) USP3 proteins. ZnF zinc-finger ubiquitin-binding domain, USP ubiquitin-specific protease domain. **i** Immunoblot for USP3$^{wt}$ and USP3$^{C168S}$ proteins and HA-MYCN from HEK293T cells. **j** Immunoblot for Flag-USP3$^{wt}$ or Flag-USP3$^{C168S}$ after immunoprecipitation of HA-MYCN. Vector indicates empty vector control. **k** HEK293T cells expressing USP3$^{wt}$ and USP3$^{C168S}$ were treated with cycloheximide (CHX) and subjected to immunoblot for MYCN and Flag-USP3. **l** HEK293T cells expressing HA-MYCN, His-Ubiquitin, USP3$^{wt}$, and USP3$^{C168S}$ or vector control were subjected to HA-MYCN immunoprecipitation and immunoblot for ubiquitination. The dot (a measure of center for the error bars) in each row represents the mean hazard ratio, while the error bars represent the 95% confidence intervals for the presented mean hazard ratio in **c**. The line in the middle of the box plot represents the median expression value, and the upper/lower bounds of the boxes represent the interquartile range of all expression values. The whiskers represent 1.5× the interquartile range of all expression values, from the upper/lower bounds of the box in **d** and **e**. Data are representative of three independent experiments with similar results in **f**, **g**, and **i–l**.

---

the binding of ALYREF to MYCN at AA 281–345, enhances the affinity of MYCN to the chromatin. Genome-wide intergenic binding of ALYREF might also establish long-range interactions by chromatin looping or chromatin remodeling leading to gene regulation. Our data also suggest that ALYREF binding to MYCN AA 281–345 may interfere with other MYCN cofactors known to bind in close proximity to this region (HAUSP[43], PA2G4[38], LSD1[55], and WDR5[56]). It is known that USP28[57] and Aurora-A[58] bind to MYC proteins via the same motif which is recognized by FBXW7 at MYC Box I, and that Aurora-A competes with FBXW7 for binding, thereby effectively lowering the affinity of MYCN for FBXW7. The interaction between MYCN and ALYREF may limit or facilitate secondary interactions between MYCN and USP7[43] or PA2G4[38] to stabilize MYCN or modulate transcriptional responses with LSD1[55] or WDR5[56]. We conclude that the observed ALYREF binding is biologically significant in the context of malignant cells, however, the exact role of ALYREF in regulating MYCN-dependent or -independent gene transcription in neuroblastoma remains largely undetermined.

Direct pharmacological approaches to inhibition of MYCN have been notoriously difficult. Alternatively, MYCN upstream positive regulators, especially enzymes, may be better targets for treatment. Indeed, it has been demonstrated that inhibitors of E3 ubiquitin ligases AURKA[59–61] and PLK1[62] promote MYCN protein degradation. Deubiquitinating enzymes (DUBs) are increasingly recognized as attractive targets for therapeutic intervention as inhibition of their activity is a more direct approach than activating E3 ligases. To date, only two DUBs have been implicated in negative regulation of MYCN, USP28[57], and USP7[43]. The USP7 specific DUB-inhibitor P22077 already has been introduced as a potential therapeutic agent for *MYCN*-driven neuroblastoma[43]. Destabilizing MYCN represents an emerging therapeutic strategy, however, there is still a need to identify novel molecular mechanisms allowing the design of inhibitors that more potently and selectively reduce MYCN protein levels and activity. In our study, we identified USP3 as an MYCN DUB, further expanding our knowledge of the diversity of MYCN protein homeostasis. Here we show that USP3 cleaves K-63-linked polyubiquitin chains off MYCN and can regulate MYCN protein turnover. Although the K-48 linkage on its own acts as a mark for proteasomal degradation of MYCN, the function of K-63 chains in MYCN protein degradation is less clear[40]. We argue that downregulation of USP3 through genetic inhibition of the ALYREF-MYCN protein complex disturbed MYCN protein homeostasis that also changed the abundance of K-48-linked polyubiquitin chains on MYCN and led to proteasomal

degradation. Interestingly, previous studies have shown that K63-ubiquitination of MYC by Huwe1[63], HectH9[64], or Skp2[65] stimulates the transcriptional activity of MYC in addition to regulating its turnover.

*ALYREF* expression is dysregulated in several cancer types[22,23]. Studies have implicated ALYREF as a regulator of cancer cell growth, since depletion of ALYREF results in a significant decrease in cell proliferation[24]. Furthermore, ALYREF knockdown caused a significant decrease in cancer cell migration[23]. Our in vitro and in vivo data strongly supports the role of ALYREF as a co-driver for oncogenesis in neuroblastoma. We also suggest that the regulatory relationship described in this work between ALYREF and MYCN may extend to other human cancer types with high MYC/MYCN dependency (medulloblastoma, retinoblastoma, hematologic malignancies, glioblastoma, ovarian cancer, Wilms' tumor, neuroendocrine prostate cancer). High *ALYREF* expression in neuroblastoma tumor tissue had prognostic significance, which was independent of *MYCN*-amplification, therefore we argue that ALYREF may have MYCN-independent oncogenic effects. Despite considerable analysis carried out in this study on the ALYREF and MYCN regulatory circuit, it is not yet fully clear how *MYCN*-promoted oncogenic levels of ALYREF reprogram cells to help the cancer state.

Our study extends the knowledge regarding genes crucial for tumor progression in 17q21-ter gain neuroblastoma patients, and indicates that the ALYREF-USP3-MYCN regulatory complex is valuable as a potential target for the treatment of neuroblastoma by exploiting the enforced ALYREF addiction present in *MYCN*-amplified cells.

## Methods

### Experimental models and subject details
*Cell culture.* Neuroblastoma cell lines, SHEPMYCN3, SK-N-AS, SK-N-DZ, SK-N-FI, SK-N-BE(2)C, SH-EP, SH-SY5Y, LAN-1, IMR-32, NBLS, and embryonic kidney cells HEK293T were maintained in Dulbecco's modified Eagle's medium (DMEM) (Life technologies Australia, VIC, Australia) with 10% fetal calf serum (FCS) (Life Technologies). CHP-134 and Kelly cells were cultured in RPMI media (Life Technologies) with 10% FCS. MRC5 and WI-38 normal human fibroblasts were grown in alpha-minimum essential media (MEM) (Life Technologies) supplemented with 10% heat-inactivated FCS. Neuroblastoma cell line SHEPMYCN3 was kindly provided by Professor Jason Shohet (Texas Children's Cancer Center, Houston, TX, USA). Neuroblastoma cell lines, SK-N-BE(2)C, SH-EP, and SH-SY5Y cells were provided by Barbara Spengler (Fordham University, New York, NY). Neuroblastoma cell lines, IMR-32 and SK-N-FI as well as HEK293T cells were obtained from the American Type Culture Collection (ATCC) (Manassas, VA, USA). The Lenti-X™ 293T viral packaging cell line was purchased from Scientifix (South Yarra, Victoria, Australia) and maintained in DMEM. Neuroblastoma cell lines, Kelly, CHP-134, SK-N-DZ, and SK-N-AS cells were obtained from the European Collection of Cell Cultures through Sigma (Sigma, Sydney, Australia).

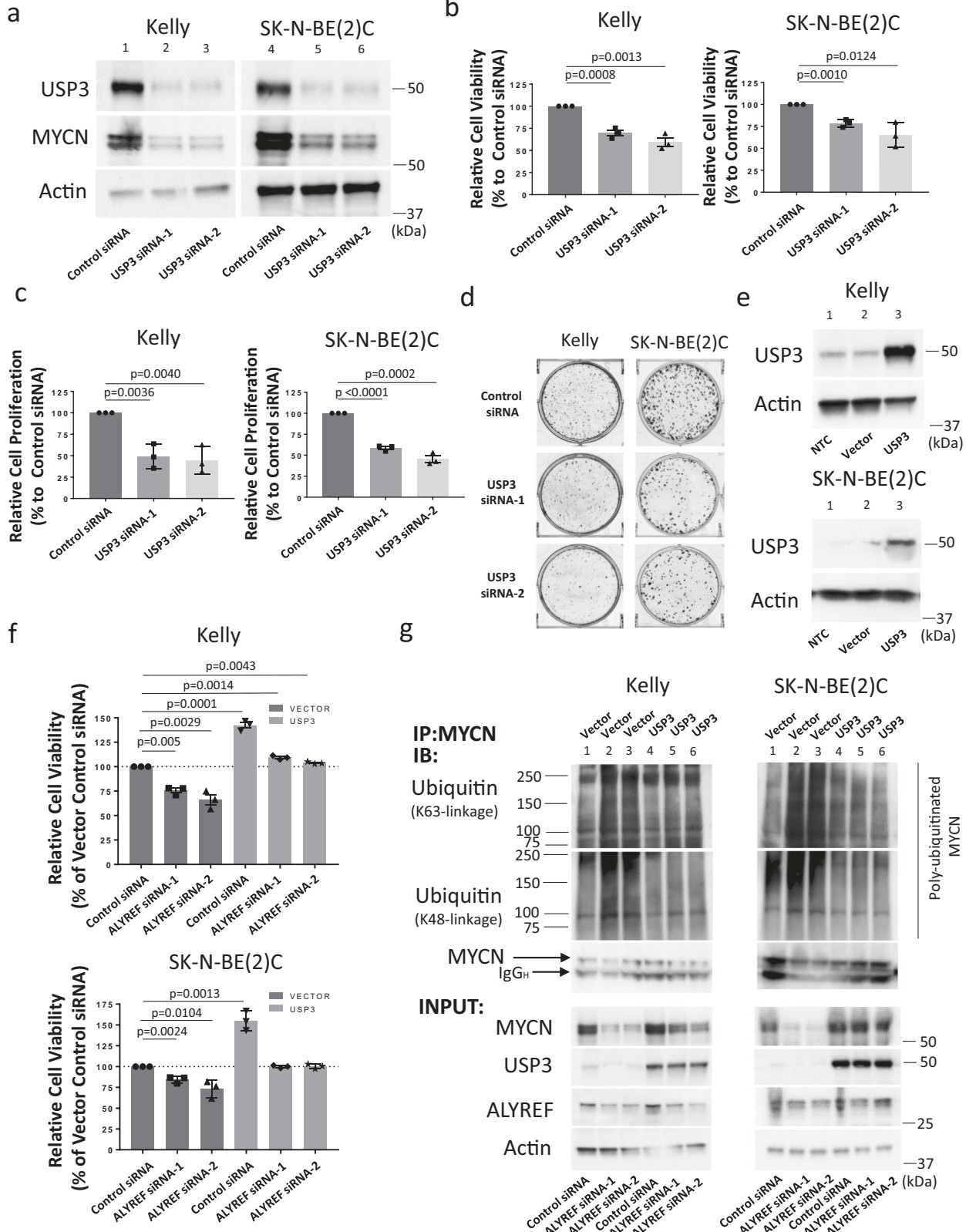

Neuroblastoma cell line NBLS was kindly provided by Prof. Susan L. Cohn (Northwestern University, Chicago, IL, USA). Neuroblastoma cell line LAN-1 was kindly provided by Dr. John Maris (Children's Hospital of Philadelphia, Philadelphia, USA). MRC5 and WI-38 normal human fibroblasts were purchased from ATCC. All cell lines used were authenticated by Cell Bank Australia (Westmead, NSW, Australia), free from mycoplasma, and cultured at 37 °C/5% $CO_2$ in a humidifier incubator. All cells were freshly thawed from initial seeds and were not cultured for more than 2 months.

*Establishment of doxycycline-inducible control shRNA and ALYREF shRNA-expressing neuroblastoma cell lines.* To generate control and stable ALYREF-knockdown cells, SK-N-BE(2)C and Kelly cells were infected with pTRIPZ non-targeting control vector (RHS5087-EG10189) or pTRIPZ encoding an ALYREF-specific shRNA (5′-AGAGTTCCTGAATATCGGC-3′) purchased from Dharmacon (Millenium Science Australia, VIC, Australia). Lentiviral vectors were packaged in HEK293 cells according to the manufacturer's protocol. 48 h later, the viral supernatant was incubated with polybrene (Santa Cruz

**Fig. 7 USP3 is responsible for ALYREF-driven MYCN ubiquitination. a** Immunoblot analysis of MYCN and USP3 expression in Kelly and SK-N-BE(2)C cells following siRNA-mediated USP3 knockdown for 48 h. **b** Cell viability ($n = 3$ per group) (**c**) and proliferation ($n = 3$ per group) analysis of Kelly and SK-N-BE(2)C cells transfected with USP3 siRNA-1, USP3 siRNA-2, or Control siRNA for 96 h. Two-sided unpaired Student's $t$-tests were performed to derive $p$-values. Differences in cell growth were compared to Control siRNA expressing cells. **d** Kelly and SK-N-BE(2)C cells transfected with USP3 siRNA-1, USP3 siRNA-2, or Control siRNA were grown for 12 days (Kelly) or 10 days (SK-N-BE(2)C), followed by colony formation assay. **e** Immunoblot analysis of USP3 expression in stable Kelly and SK-N-BE(2)C cells overexpressing USP3 (USP3) or Vector control (Vector). NTC, non-transduced control cells. **f** Stable Kelly and SK-N-BE(2)C cells overexpressing USP3 (USP3) or Vector control (Vector) were co-transfected with ALYREF siRNA-1 or ALYREF siRNA-2 or Control siRNA, followed by cell viability measurements ($n = 3$ per group) at 72 h. Two-sided unpaired Student's $t$-tests were performed to derive $p$-values. Differences in cell viability were compared to Vector Control siRNA expressing cells. **g** Stable Kelly and SK-N-BE(2)C cells overexpressing USP3 (USP3) or Vector control (Vector) were co-transfected with ALYREF siRNA-1, or ALYREF siRNA-2 or control siRNA, followed by treatment with MG132 for 4 h. Cells were then subjected to endogenous MYCN immunoprecipitation and immunoblot analyses for K-63 and K-48-linked ubiquitination. Respective $p$-values ($p$) are displayed. Comparisons were not significant unless otherwise noted. Data are representative of three independent experiments with similar results in **a**, **d**, and **e**. Data are representative of two independent experiments with similar results in **g**. Data are shown as mean ± s.e.m. (error bars) and representative of three independent experiments in **b**, **c**, and **f**.

Biotechnology, Santa Cruz, CA) and added to SK-N-BE2C and Kelly cells. 2 µg/ml puromycin (Sigma) was used to select positive clones.

*Generation of USP3-overexpressing neuroblastoma cell lines.* To generate stable USP3-overexpressing neuroblastoma cells, SK-N-BE(2)C and Kelly cells were infected with Precision LentiORF encoding control ORF or Precision LentiORF encoding USP3 ORF lentiviral particles (OHS5899–202618161 and OHS5900–202620492) purchased from Dharmacon (Millenium Science Australia). The viral particles were incubated with polybrene (Santa Cruz Biotechnology) and added to SK-N-BE(2)C and Kelly cells. 7.5 µg/ml blasticidin (Sigma) was used to select positive clones.

*In vivo mouse experiments.* Animal experiments were approved by the Animal Care and Ethics Committee of UNSW Australia (ACEC#18/113B), and the care of the animals was in accord with institutional guidelines. Mice were held under PC2 conditions in individually ventilated caging with a maximum of 6 or a minimum of 2 mice per cage. Dimensions for mice cages: 369 × 156 × 132 mm, with a floor area of 440 cm². The mice were maintained in a protected and controlled environment. The animal facility was barrier protected with the air HEPA filtered and the room maintained at positive pressure and at a temperature 22 ± 1 °C. Sterile feed and water were provided ad libitum. The light in the animal facility was on a 12 h cycle. All experimental work was conducted in biosafety or cytotoxic cabinets to further protect the animals from the microbiological threat. Female Balb/c nude mice aged 5–6 weeks were injected subcutaneously with $2 \times 10^6$ doxycycline-inducible ALYREF shRNA BE(2)C cells into the right flank. When engrafted tumors reached 5 mm size in diameter, tumor-bearing mice were randomized into two groups: vehicle (DMSO)-treated and doxycycline (DOX)-treated mice. Mice with undetectable engraftment were excluded. Mice were fed with 5% sucrose water with or with doxycycline (Sigma) at 2 mg/mL. Tumor development was monitored and tumor volume was calculated using (length × width × height)/2. Mice were culled when tumor volume reached 1 cm³, and tumor tissues were snap-frozen and analyzed by immunoblotting for ALYREF, MYCN, USP3, and actin protein expression.

*Microarray analysis on TH-MYCN$^{+/+}$ ganglia tissues.* Ganglia from TH-MYCN$^{+/+}$ mice and wild-type controls were collected at 2, 4, and 6 weeks of age. RNA was extracted from the cells with PureLink RNA Kit (Life Technologies), and genome-wide differential gene expression was examined using Agilent SurePrint G3 Mouse GE 8×60K Microarrays. The microarray data were analyzed in R [http://www.r-project.org/] and normalized using GenePattern software (version 3.2.3 Broad Institute) with the AgilentToGCT and LimmaGP modules (version 19.3) available at https://pwbc.garvan.unsw.edu.au/gp[35].

*Patient tumor genome/gene expression/survival analyses.* Whole-genome sequencing (WGS) data, containing normalized coverage and ploidy calls for genomic segments in 135 clinically annotated primary neuroblastoma tumors (TARGET neuroblastoma cohort), were obtained through the TARGET data matrix (https://ocg.cancer.gov/programs/target/data-matrix) and further processed using the R statistical language and R Studio. Average ploidy values were calculated in 10 kb bins across chromosome 17 using the Genomic Ranges R package (v1.40.0), wherein an average ploidy ($n$), $n \leq 1$ was considered a loss, $1 \leq n \leq 3$ was considered diploid and $n \geq 3$ considered to be a gain[25]. Average ploidy across the 17q21-ter region was then calculated in each sample and classified according to the above strata. We utilized matched clinical annotations containing event-free and overall survival data to construct Kaplan–Meier survival curves of the different molecular subgroups using the survminer (v0.4.6) and survival R packages (v2.42.1). RNA-seq data, in the form of gene-level counts, for 90 samples, which had paired WGS data (aforementioned), were also obtained from the TARGET data matrix. We performed differential gene expression testing between 17q21-ter gain vs diploid as

well as *MYCN*-amplified vs non-amplified samples using the DESeq2 R package (v1.22.0)[66] and filtered for genes passing an adjusted $p$-value threshold of <0.05 and log2 fold-change threshold being ≥ the lower quartile of all positive log2foldchange values passing the adjusted $p$-value threshold. Correlation of gene expression between genes of interest as well as with gene copy number was analyzed using Pearson correlation and resultant $p$-values were adjusted using the Bonferroni method, followed by filtering using an adjusted $p$-value threshold of <0.05 and $r > 0$. Using the survival R package (v2.42.1) univariate Cox proportional hazards (CoxPH) regression models for genes of interest were performed and then filtered based on a hazard ratio (HR) threshold of >1 and individual log-rank test $p$-value < 0.05.

RNA-seq data, expressed in reads per million (RPM) for each gene, for 498 clinically annotated primary neuroblastoma samples (SEQC neuroblastoma cohort) were obtained from the gene expression omnibus (GEO) with the accession GSE62564[31]. Microarray data, for each gene, for 476 neuroblastoma samples (Kocak neuroblastoma cohort), were downloaded from R2 platform (http://r2.amc.nl). We utilized matched clinical annotations containing event-free and overall survival data to construct Kaplan–Meier survival curves of subgroups dichotomized by median gene expression using the survival R package (v2.42.1). Using the same R package (v2.42.1), we ran either univariate or multivariate CoxPH regression models for genes of interest, regressing covariates associated with neuroblastoma prognosis such as advanced disease stage (3 and 4), age at diagnosis (>18 months), and *MYCN*-amplification status. All statistical tests concerning Kaplan–Meier analyses were done utilizing log-rank tests, adjusted using the Bonferroni method for multiple hypotheses testing where appropriate.

**Method details**

*siRNA and plasmid DNA transfections.* For siRNA-mediated knockdown, the indicated cell lines were transfected with 20 nM of predesigned siRNA duplex oligos (MYCN siRNA#1: 5′-CGTGCCGGAGTTGGTAAAGAA-3′; MYCN siRNA#2: 5′-CAGCGCGTTTCTTCTCCTGTA-3′; ALYREF siRNA#1: 5′-TTGCTGAATTTGGAACGCTGAA-3′; ALYREF siRNA#2: 5′- CAGAGGTGG CATGACTAGAAA-3′) synthesized by Qiagen (VIC, Australia). ON-TARGETplus Human USP3 siRNA (LQ-006078-00-0020) synthesized by Dharmacon (Millenium Science). Non-targeting pool siRNA was used as siControl and purchased from Dharmacon (Millenium Science).

For overexpression experiments, the indicated cell lines were transfected with or 3–15 µg (for T25 and T75 flasks respectively) pCMV6-ALYREF-Myc/Flag (Origene Technologies, Rockville, MD, USA); pCMV6-Empty (Origene Technologies); pCMV3Tag3A-USP3$^{wt}$-Flag (GeneScript, Integrated Sciences, NSW, Australia); pCMV3Tag3A-USP3$^{C168S}$-Flag (GeneScript); pCMV3Tag3A-Empty (GeneScript); pCMV-His-Ubiquitin (GeneScript), pcDNA3.1-MYCN-HA or pcDNA3.1-Empty plasmid DNA. All HA-MYCN deletion mutants were generated in pcDNA3.1 vectors and provided by Professor Wei Gu. The catalytically defective enzyme mutant for USP3 with a serine residue substituting for cysteine (C168S) was generated by GeneScript from pCMV3Tag3A-USP3$^{wt}$-Flag by PCR-based site-directed mutagenesis. Cells were transfected between 24 and 96 h depending on the experimental requirements. Transfections were performed using Lipofectamine 2000 (Life Technologies) following the manufacturer's protocol.

*RNA isolation and quantitative real-time PCR.* Total RNA was extracted using the PureLink RNA kit (Life Technologies) according to the manufacturer's protocol. 1 µg total RNA was reverse-transcribed using the Tetro cDNA synthesis kit (Life Technologies). Gene expression was verified using Power SYBR Green mix (Applied Biosystems, Life Technologies) performed on ABI7500 thermo-cycler (Applied Biosystems, Life Technologies) with a standard protocol. Differential gene expression was measured using the log2ΔΔCt analysis. All mRNA expression levels were normalized to glyceraldehyde-3-phosphate dehydrogenase (GAPDH) or Beta-

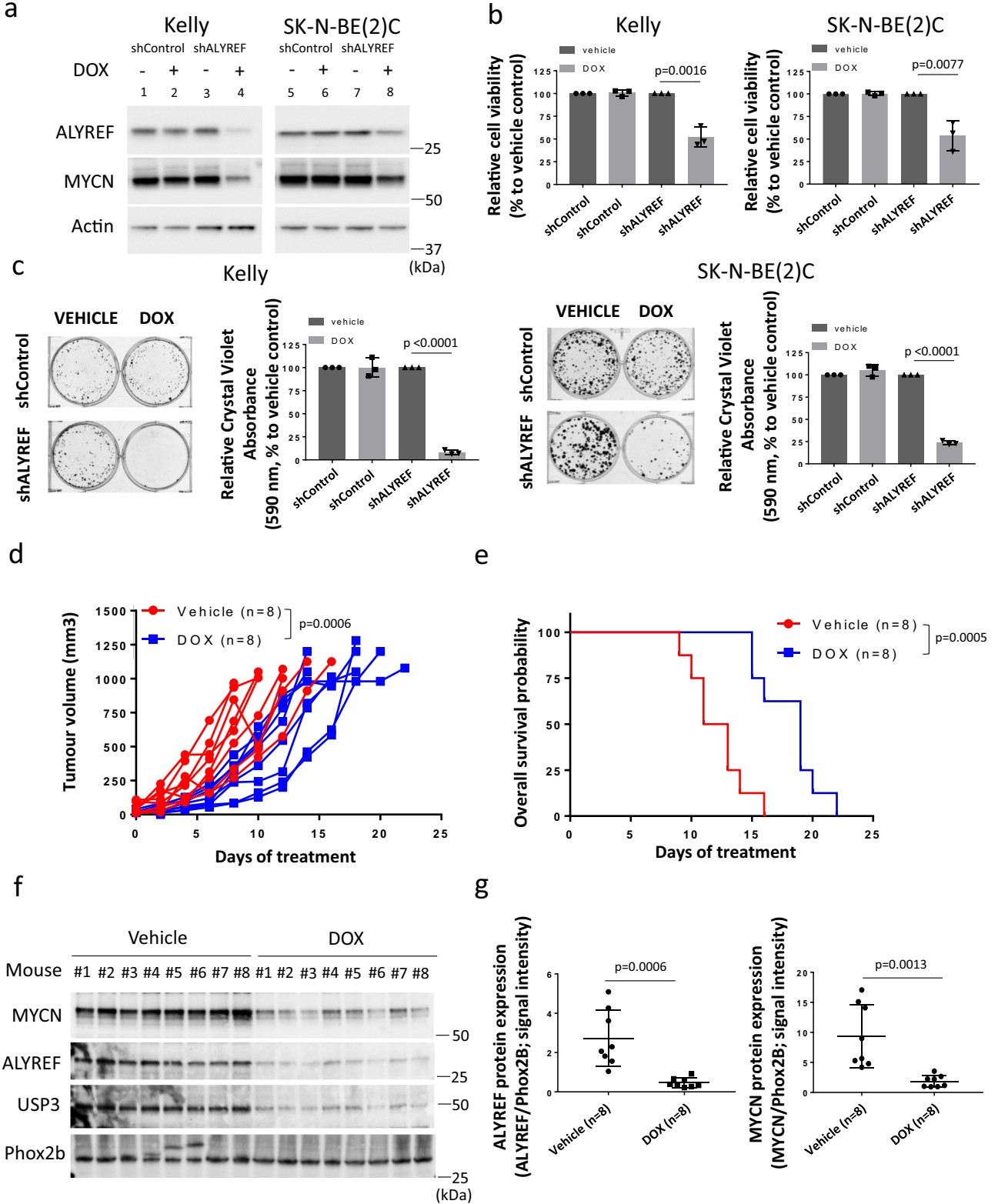

2-Microglobulin (B2M). Primer sequences (Supplementary Table 5) were obtained from Integrated DNA Technologies (NSW, Australia).

*Immunoprecipitation.* For endogenous MYCN or ALYREF immunoprecipitation assays, cells were lysed in cold NP40 buffer (Sigma) supplemented with protease inhibitor (Sigma). For endogenous USP3 immunoprecipitation assays, cells were lysed in cold BC100 buffer (Sigma) supplemented with protease inhibitor (Sigma). 5% of the cell extract was saved as the input, and the rest (750 μg total protein) was incubated with either 10 μg MYCN-specific antibody (Merck Millipore, VIC,

Australia), 10 μg ALYREF-specific antibody (D3R4R; Cell Signaling Technologies, Beverly, MA, USA), or with 10 μg control IgG antibody (mouse from Santa Cruz Biotechnology and rabbit from Cell Signaling) and A/G PLUS agarose beads (Santa Cruz Biotechnology) at 4 °C overnight. After three washes with the lysis buffer, the bound proteins were eluted by boiling with SDS sample buffer. Bound proteins were resolved by SDS-PAGE.

For immunoprecipitation of ectopically expressed Flag and HA-tagged proteins, HEK293T human embryonic cells were transiently co-transfected with an HA-MYCN-expression pcDNA3.1 construct, together with an empty vector or

**Fig. 8 ALYREF is required for the growth and tumorigenicity of *MYCN*-amplified neuroblastoma cells.** Kelly and SK-N-BE(2)C cells expressing ALYREF shRNA (shALYREF) were treated with doxycycline (DOX; 2 μg/ml) for 72 h, then subjected to immunoblot analysis of MYCN and ALYREF expression (**a**) and cell viability (**b**) measurements (*n* = 3 per group). Two-sided unpaired Student's t-tests were performed to derive *p*-values. Differences in cell viability were compared to the vehicle control shRNA (vehicle). **c** Kelly and SK-N-BE(2)C cells expressing ALYREF shRNA (shALYREF) or control shRNA (shControl) were treated with doxycycline (DOX) for 12 days (Kelly) or 10 days (SK-N-BE(2)C), followed by colony formation assay (*n* = 3 per group). Two-sided unpaired Student's *t*-tests were performed to derive *p*-values. Differences in colony formation were compared to the vehicle control shRNA (vehicle). SK-N-BE(2)C cells expressing ALYREF shRNA (shALYREF) were xenografted into nude mice. Once tumors reached 5 mm in diameter, the mice were divided into doxycycline (DOX) and vehicle control (Vehicle) subgroups and fed with 10% sucrose water, with or without 2 mg/ml doxycycline. **d** Tumor volume was taken from each mouse (*n* = 8 per treatment group), starting from day 0 post-treatment, until the tumor reached 1000 mm³. The effect of DOX on tumor progression was evaluated using two-way ANOVA **e** Kaplan–Meier survival curves showed the probability of overall survival of the mice (*n* = 8). A two-sided long-rank test was used to obtain *p*-value. Immunoblot (**f**) and protein densitometry (**g**) analysis of tumor samples (*n* = 8) from SK-N-BE(2)C tumor xenografts, analyzed for ALYREF, USP3, and MYCN expression. Phox2B expression was used as a neuroblastoma marker and loading control. Two-sided unpaired Student's *t*-tests were performed to derive *p*-values. Differences in ALYREF and MYCN protein expressions were compared to the vehicle-treated ALYREF shRNA subgroup. Respective *p*-values (*p*) are displayed. Comparisons were not significant unless otherwise noted. Data is representative of three independent experiments with similar results in **a** and **c**. Data are shown as mean ± s.e.m. (error bars) and representative of three independent experiments in **b** and **c** and 8 mice per treatment group in **d**. In vivo data are representative of two experiments in **f** and quantification is shown as mean ± s.e.m. (error bars) of each treatment group in **g**.

Flag-ALYREF-expression pCMV6 construct or Flag-USP3-expression pCMV3Tag3A construct for 24 h. Cells were lysed in cold NP40 or BC100 buffer supplemented with protease inhibitor (Sigma), respectively. 5% of the cell extract was saved as the input, and the rest (500 μg total protein) was incubated with either 15 μg of Flag (DYKDDDDK)-specific antibody (9A3; Cell Signaling Technologies), 15 μg HA-specific antibody (3F10; Sigma) or with 15 μg control IgG (Santa Cruz Biotechnology) and A/G PLUS agarose beads (Santa Cruz Biotechnology) at 4 °C overnight. After three washes with the lysis buffer, the bound proteins were eluted by boiling with SDS sample buffer.

For HA-MYCN deletion mutant experiments, pcDNA3.1-HA-MYCN plasmid or the mutant constructs were co-transfected with the construct expressing ALYREF for 24 h, and the cells were lysed in NP40 buffer (Sigma). 5% of the cell extract was saved as the input, and the rest (500 μg total protein) were incubated with either 15 μg of Flag (DYKDDDDK)-specific antibody (9A3; Cell Signaling Technologies) or with 15 μg control IgG (Santa Cruz Biotechnology) and A/G PLUS agarose beads (Santa Cruz Biotechnology) at 4 °C overnight. After three washes with the lysis buffer, the bound proteins were eluted by boiling with SDS sample buffer. Bound proteins were resolved by SDS-PAGE.

*Western blot analysis.* Cell pellets were lysed with NP40 or BC100 buffer (Sigma) freshly supplemented with protease inhibitor cocktail (Sigma). Protein lysate was standardized using the BCA protein quantitation assay kit as per manufacturer's instructions (ThermoFisher Scientific, Waltham, MA, USA), and 20–40 μg whole protein lysates were resolved on either 10.5% or 10–14% Tris-HCl Criterion gels (Bio-Rad, Gladesville, NSW, Australia). Nitrocellulose membranes (GE Healthcare, Rydalmere, NSW, Australia) were blocked with 5% (wt/vol) nonfat dry milk in Tris-buffered saline with Tween-20 (20 mM Tris-HCl (pH 7.6), 137 mM NaCl, 0.1% Tween-20) and incubated overnight at 4 °C with the following primary antibodies: ALYREF (D3R4R; 1:1000; Cell Signaling Technologies); MYCN (B84B; 1:1000; Santa Cruz Biotechnologies); cMYC (D84C12; 1:500; Cell Signaling Technologies); β-actin (AC-15; 1:10,000; Sigma), GAPDH (G-9; 1:1000; Santa Cruz Biotechnology)**;** Flag/DYKDDDDK tag (9A3; 1:1000; Cell Signaling Technologies); K-48 linkage-specific ubiquitin (EP8589; 1:1000; AbCam, VIC, Australia); K-63 linkage-specific ubiquitin (HWA4C4; 1:1000; Invitrogen, ThermoFisher Scientific); ubiquitin (P4D1; 1:500; Cell Signaling Technologies); HA (3F10; 1:1000; Sigma); Topoisomerase I (1:10,000; Novus Biologicals, In vitro Technologies, VIC, Australia) and USP3 (1:1000; Invitrogen, ThermoFisher Scientific). Appropriate horseradish peroxidase-conjugated secondary antibodies (1:3000; Santa Cruz Biotechnologies and Merck Millipore) were diluted in Tris-buffered saline with 0.1% Tween-20 and membranes were probed at room temperature for 2 h. Immunoblots were visualized with Clarity ECL Chemiluminescence reagents (Bio-Rad). Densitometry of protein expression was measured using Quantity One software (Bio-Rad) and each protein expression band was normalized to loading controls. Uncropped immunoblot scans are provided as Supplementary Information.

*Isolation of nuclear and cytoplasmic extract.* The nuclear and cytoplasmic extracts were prepared using a NE-PER Nuclear Cytoplasmic Extraction Reagent kit (ThermoFisher Scientific) according to the manufacturer's instruction. Briefly, cells were washed twice with cold PBS and centrifuged at 500×g for 3 min. The cell pellet was suspended in 200 μl of cytoplasmic extraction reagent I by vortexing. The suspension was incubated on ice for 10 min followed by the addition of 11 μl of a second cytoplasmic extraction reagent II, vortexed for 5 s, incubated on ice for 1 min, and centrifuged for 5 min at 16,000×g. The supernatant fraction (cytoplasmic extract) was transferred to a pre-chilled tube. The insoluble pellet fraction, which contains crude nuclei, was resuspended in 100 μl of nuclear extraction reagent by vortexing for 15 s and incubated on ice for 10 min, then centrifuged for 10 min at

16,000×g. The resulting supernatant, constituting the nuclear extract, was used for the subsequent experiments along with the cytoplasmic extract.

*MYCN stability analysis.* SK-N-BE(2)C cells were transfected with pCMV6-ALYREF plasmid DNA for 48 h or stable ALYREF knockdown SK-N-BE(2)C and Kelly cells were induced with doxycycline for 72 h, then treated with 100 μg/ml cycloheximide (Sigma) for 0–60 min. Cells were collected at 15 min treatment intervals and protein lysates were analyzed by Western blot to measure changes in MYCN protein stability.

HEK293T cells were co-transfected with pCMV3Tag3A-Flag-USP3^WT or pCMV3Tag3A-Flag-USP3^C168S mutant constructs along with pcDNA3.1-HA-MYCN plasmid DNA for 24 h, then treated with 100 μg/ml cycloheximide (Sigma) for 0–60 min. Cells were collected at 15 min treatment intervals and protein lysates were analyzed by western blot to measure changes in MYCN protein stability.

*Ubiquitin assays.* Cells were exposed to the proteasomal inhibitor MG132 (Sigma) at 20 μM concentration[42]. Four hours post MG132 treatment, cells were lysed with denaturing lysis buffer (6 M guanidine-HCl, 50 mM sodium phosphate, pH 8.0, 500 mM NaCl, 5 mM imidazole). 5% of the sample was saved to serve as an input. The rest of the sample (750 μg total protein) was diluted in BC100 buffer (Sigma) freshly supplemented with protease inhibitor cocktail (Sigma) and subjected to immunoprecipitation with either 5 μg MYCN-specific antibody (Merck Millipore), or with 5 μg control mouse IgG antibody (Santa Cruz Biotechnology) and A/G PLUS agarose beads (Santa Cruz Biotechnology) at 4 °C overnight. Prior to immunoprecipitation, the beads were washed three times with denaturing wash buffer (8 M urea, 50 mM sodium phosphate, pH 8.0, 500 mM NaCl). After overnight incubation at 4 °C, beads were washed five times with cold BC100 buffer (Sigma) and once with cold sodium chloride solution (150 mM). Proteins were eluted in Laemmli sample buffer (62.5 mM Tris-HCl, pH 6.8; 25% glycerol; 2% SDS; 0.01% Bromophenol Blue) supplemented with β-mercaptoethanol (Bio-Rad) at a final concentration of 5%, and separated by electrophoresis using 10.5% or 10–14% Tris-HCl Criterion gels (Bio-Rad).

*Chromatin-immunoprecipitation (ChIP) assays and double-antibody ChIP assays.* Preparation of DNA from SK-N-BE(2)C and Kelly cells for ChIP assay was performed using the Chromatin-Immunoprecipitation Assay kit (Merck Millipore) according to the manufacturer's instructions. ChIP assays were performed with mouse anti-MYCN antibody (B84B; Santa Cruz Biotechnology); anti-ALYREF antibody (11G5; ImmunoQuest) or control mouse IgG antibody (Santa Cruz Biotechnology). DNA was purified using a MiniElute PCR Purification Kit (Qiagen). Real-time PCR was performed with primers designed to cover the regions of the *ALYREF* and *USP3* genes containing MYC-responsive or ALYREF-responsive motifs or remote negative control regions. Fold enrichment of the *ALYREF* or *USP3* genes containing the binding regions by the anti-MYCN or anti-ALYREF antibody was calculated by dividing the PCR product from this region by the PCR product from the negative control region, relative to the input.

For double-antibody ChIP assays, we performed sequential pull-downs, first with anti-MYCN antibody (B84B; Santa Cruz Biotechnology) followed by anti-ALYREF antibody (11G5; ImmunoQuest) using appropriate IgG controls.

*Chromosome conformation capture (3C) assay.* For 3C assay[67], nuclei were extracted from fixed SK-N-BE(2)C cells as described in the chromatin-immunoprecipitation protocol. BamH1 and HindIII (Promega Australia, NSW, Australia) restriction enzymes were selected based on in silico analysis to cut DNA. The digested nuclei were then treated with T4 DNA ligase (ThermoFisher

Scientific) to ligate DNA fragments that interacted with each other. DNA fragments after ligation were purified using phenol/chloroform/isoamyl alcohol. Then PCR was performed for 35 cycles using the following PCR conditions: 94 °C for 10 min; and thermocycling at 94 °C for 30 s, 60 °C for 30 s and 72 °C for 30 s; following by 72 °C 15 min. PCR product was loaded on a 1.5% agarose gel with SYBR Safe (Life Technologies), and images were taken using a ChemiDoc Imaging System (Bio-Rad). PCR samples from the 3C assay were also run on a DNA High Sensitivity (HS) chip using the Agilent 2100 Bioanalyzer (Agilent Technologies Australia, VIC, Australia) according to the manufacturer's instructions. Uncropped PCR gel scans are provided as Supplementary Information.

*Cell proliferation assay.* Cell proliferation was measured using the BrdU ELISA kit (Roche, Australia) according to the manufacturer's instructions. Cells were grown in a medium containing 5 µg/ml BrdU for 4 h and fixed in 4% paraformaldehyde. DNA was denatured, and cells were permeabilized in 2 N HCl with 0.5% Triton X-100 (Sigma) and then blocked with 5% BSA in PBS. Anti-BrdU (1:200) was added following the manufacturer's protocol. After washing with 5% BSA in PBS, the cells were incubated with Alexa-Fluor-594-conjugated anti-mouse-IgG (1:100). Changes in cell proliferation were calculated from the absorbance readings at 370 nm (490 nm reference wavelength) on the Benchmark Plus microplate reader (Bio-Rad).

*Colony formation.* 200 SHEPMYCN3 cells transfected with control siRNA, ALYREF siRNA-1, or ALYREF siRNA-2 were seeded in 6-cm$^2$ plates and treated with and without doxycycline (Sigma) for 12 days. 250 SK-N-BE(2)C and 500 Kelly cells expressing control siRNA, USP3 siRNA-1, or USP3 siRNA-2 were seeded in 6-cm$^2$ plates and grown for 10 and 12 days, respectively. 250 SK-N-BE(2)C and 500 Kelly cells expressing shALYREF or control shControl constructs were seeded in 6-cm$^2$ plates and kept under doxycycline treatment for 10 and 12 days, respectively, to allow colonies to form. Colonies were fixed and stained with crystal violet solution (Sigma) then washed with water to remove the unincorporated stain. Cells were photographed and colony formation was quantified from crystal violet absorbance readings at 590 nm on the Benchmark Plus microplate reader (Bio-Rad). Uncropped colony formation assay scans are provided as Supplementary Information.

*Cell viability assay.* Cell viability was measured using the Alamar Blue assay (Life Technologies) according to the manufacturer's protocol. Signals were quantitated on a VICTOR$^2$ Multilabel counter (Perkin Elmer Australia) at an excitation wavelength of 560 nm and an emission wavelength of 590 nm.

*Determination of cell death.* SHEPMYCN3 neuroblastoma cells ($5 \times 10^3$ cells per well) were seeded in 96-well plates, transfected with control siRNA, ALYREF siRNA-1, or ALYREF siRNA-2 for 16–48 h with and without doxycycline. The mitochondrial membrane potential of cells was determined using a MitoProbe DilC1(5) Assay Kit (Life Technologies). The cells were incubated with DilC1(5) working solution (50 nM/well) for 30 min, and the fluorescence of DilC1(5) was measured at 630-nm excitation and 670-nm emission wavelengths using VICTOR Multilabel Counter (Perkin Elmer). Dead cells were determined by SYTOX Green staining (Life Technologies). The cells were incubated with SYTOX Green working solution (30 nM/well) for 30 min, and the fluorescence of SYTOX Green was measured at 490-nm excitation and 520-nm emission wavelengths using VICTOR Multilabel counter (Perkin Elmer).

*ChIP-seq analysis.* SK-N-BE2C cells were subjected to chromatin-immunoprecipitation with a mouse anti-ALYREF antibody (11G5, Immuno-Quest) or control mouse IgG (Santa Cruz Biotechnologies). Preparation of DNA from SK-N-BE(2)C cells for ChIP-seq was performed using the Chromatin-Immunoprecipitation Assay kit (Merck Millipore) according to the manufacture's instructions. After immunoprecipitation, DNA was purified and ChIPed DNA sample libraries were prepared using TruPrep DNA Library Preparation Kit V2 for Illumina. Samples were analyzed using Illumina Hiseq X-Ten pair-end sequencing (150 bp). We also obtained several other publicly available ChIP-seq data sets of MYCN, RNA Polymerase II, BRD4, H3K27ac, H3K4me3, and H3K27me3 in the SK-N-BE(2)-C cell line, among other neuroblastoma cell lines, to complement our ALYREF ChIP-seq data (GSE80151)[36], fastq files were obtained directly from the European Nucleotide Archive (ENA) under the study accession PRJNA318044. Reads from fastq files were first quality trimmed using trimgalore (v0.4.5), followed by alignment to the human genome (GRCh38) using bowtie2 (v2.1.0)[68]. SAMtools (v1.9)[69] was then used to convert, sort, and index alignments. Reads aligned to ENCODE blacklisted regions[70] were removed using bedtools (v2.27.1). Peaks were then called using MACS2 (v2.1.1)[71] either in paired-end mode for ALYREF ChIP-seq data or single-end mode for all other data sets, with an FDR *q*-value threshold of <0.05. Fold enrichment tracks which express the relative enrichment of the ChIPed protein compared to the genomic input, were generated using MACS2 (v2.1.1) and converted to the bigwig format using BEDtools (v2.27.1)[72] for visualization on the IGV browser[73]. HOMER (v4.10.3)[74] was used to annotate peaks (promoters were considered to be −1000 bp/+100 bp from the transcription start site (TSS)) and then to perform de novo/known motif discovery on ALYREF peak regions. PWMScan[75] was used to annotate canonical (CACGTG) and non-

canonical (CANNTG) MYCN E-Boxes across the human genome (GRCh38) and were visualized on IGV alongside fold enrichment tracks.

*Multi-omic cancer cell line analyses.* We utilized public data resources produced by the Cancer Cell Line Encyclopedia (CCLE) and Project Achilles via the Cancer Dependency Map[33] portal (DepMap, 20Q1). Gene expression (transcripts per million - TPM), copy number (relative ploidy), and gene dependency (CERES) data were first obtained from DepMap and then filtered using R/R Studio (v1.1.456) for disease types/cell lines common to all data sources as well as those disease types that had >1 cell line present. Statistical tests were then performed between neuroblastoma cell lines and an aggregate of other cell lines, or *MYCN*-amplified vs non-amplified neuroblastoma cell lines, using two-sample *t*-tests.

*Statistical analysis.* All in vitro experiments for statistical analysis were performed at least three times. Data were analyzed with Prism 7 software (GraphPad) and expressed as mean ± standard error. Differences were analyzed for significance with a two-sided unpaired *t*-test for two groups or ANOVA among groups with a confidence interval (CI) of 95%. All statistical tests were two-sided. Correlation of gene expression between genes of interest as well as with gene copy number was analyzed using Pearson correlation and resultant *p*-values were adjusted using the Bonferroni method, followed by filtering using an adjusted *p*-value threshold of <0.05 and *r* > 0. All statistical tests concerning Kaplan–Meier analyses were done utilizing log-rank tests, adjusted using the Bonferroni method for multiple hypotheses testing where appropriate. No samples, mice or data points were excluded from the reported analyses, except for non-engrafted xenograft mice. Samples were not randomized to experimental groups, except for the treatment allocation in the xenograft models. Analyses were not performed in a blinded manner. For xenograft models, investigators were not blinded to group allocation.

**Reporting summary**. Further information on research design is available in the Nature Research Reporting Summary linked to this article.

## Data availability

The ALYREF ChIP-seq data have been deposited at the Gene Expression Omnibus Website with series number GSE150303. We also obtained several other publicly available ChIP-seq data sets (GSE80151) of MYCN, RNA Polymerase II, BRD4, H3K27ac, H3K4me3, and H3K27me3 to complement our ALYREF ChIP-seq data (https://www.ncbi.nlm.nih.gov/geo/query/acc.cgi?acc=GSE80151), fastq files were obtained directly from the European Nucleotide Archive (ENA) under the study accession PRJNA318044. Gene expression and relevant patient prognosis information in TARGET, SEQC, and Kocak neuroblastoma patient data sets were downloaded from R2 platform (http://r2.amc.nl). Whole-genome sequencing (WGS) data were obtained through the TARGET data matrix (https://ocg.cancer.gov/programs/target/data-matrix). RNA-seq data that had paired WGS data, were also obtained from the TARGET data matrix. RNA-seq data for the SEQC neuroblastoma cohort were obtained from the gene expression omnibus (GEO) with the accession GSE62564. We utilized public data resources produced by the Cancer Cell Line Encyclopedia (CCLE) and Project Achilles via the Cancer Dependency Map (DepMap, 20Q1) portal (https://depmap.org/portal/). Data underlying all figures and supplementary figures, uncropped and unprocessed immunoblot scans, as well as colony formation assay and PCR agarose gel pictures for all figures and supplementary figures, are provided as Source Data. All other relevant data are available from the corresponding authors on request.

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

## Acknowledgements

This work was supported by Program Grants (G.M.M.) from the National Health and Medical Research Council (NHMRC) Australia (APP1016699), Cancer Institute NSW (10/TPG/1–13), Cancer Council NSW (PG-11-06), and an Australia Postgraduate Research Award, UNSW Sydney, Australia (J.K.). This work was also supported by NHMRC Project Grant APP1125171 and Neuroblastoma Australia (G.M.M., B.B.C.). Funding from the Victorian Government Operational Infrastructure Support Scheme to St Vincent's Institute is acknowledged. We acknowledge the Steven Walter Children's Cancer Foundation for its continuous support. We also acknowledge the support from the University of New South Wales Centre for Childhood Cancer Research, UNSW Sydney, Sydney, NSW, Australia. M.W.P. is a National Health and Medical Research Council of Australia Research Fellow. HA-MYCN deletion mutants were kindly gifted by Professor Wei Gu (Institute for Cancer Genetics, College of Physicians and Surgeons, Columbia University, New York, New York, USA, Herbert Irving Comprehensive Cancer Centre, College of Physicians and Surgeons, Columbia University, New York, New York, USA, Department of Pathology and Cell Biology, College of Physicians and Surgeons, Columbia University, New York, New York, USA). We acknowledge Dr. Tzong T. Hung (Biological Resource Imaging Laboratory, the University of New South Wales, Kensington, New South Wales, Australia), Dr. Tracy L. Nero, and Michael A. Gorman (Department of Biochemistry & Molecular Biology, Bio21 Molecular Science and Bio-technology Institute, The University of Melbourne, Parkville, Victoria 3010, Australia) for their technical and administrative support.

## Author contributions

Conception and design: Z.N., B.B.C., and G.M.M. Development of methodology: Z.N., J.S., and C.M. Acquisition of data: Z.N., J.S., M.K., W.C., C.M., Y.D., A.S., J.K., and D.C. Analysis and interpretation of data: Z.N., J.S., M.K., C.M., P.V., Y.D., C.J., J.K., D.C., B.B.C., and G.M.M. Writing of the manuscript: Z.N. Review, and/or revision of the manuscript: Z.N., J.S., W.C., C.M., D.C., T.L., M.W.P., B.B.C., and G.M.M. Administrative, technical, or material support: Z.N., R.M., and M.W.P. Study supervision: Z.N., T.L., J.K., B.B.C., and G.M.M.

## Competing interests

The authors declare no competing interests.
