## [Peer Review File · Nature Communications]

REVIEWER COMMENTS

Reviewer #1 (Remarks to the Author):

The paper by Nagy et al. identifies ALYREF as a candidate 17q affected MYCN cooperative oncogene. Chromosome 17q gain has been identified as one of the most frequently recurrent chromosomal imbalances with prognostic impact but with hitherto poorly assigned functional impact. Deciphering the putative functional consequences of dosage effects resulting from 17q gain is therefore of utmost importance in order to better understand complex neuroblastoma biology and potentially also to open venues for novel therapeutic interventions. From this perspective this paper is highly interesting and illustrates the potential to use combined approaches to dissect 17q gain functionality.

The authors used an integrated analysis to identify ALYREF as candidate 17q co-driver, specifically in the context of MYCN driven tumor formation. High ALYREF expression was found to be associated with 17q21-ter gain, MYCN-amplification and poor patient prognosis in neuroblastoma. MYCN-induced increased viability and proliferation were shown to be ALYREF dependent while MYCN was shown to directly regulate ALYREF while evidence was provided that ALYREF is a critical factor in maintaining high level MYCN expression through inhibition of its protein degradation and forming a transcriptional activator complex which upregulates USP3 expression. Tumor dependency for ALYREF was also supported by in vivo experiments using modified neuroblastoma cell lines with regulable knock down constructs.

This is a very well conducted study with appropriate model systems and controls. These findings should support further efforts to test the role of ALYREF as co-driver in MYCN (or MYC) driven neuroblastomas.

Comments:

1. It is of interest that next to MYCN high level expression, also MYC impacts on ALYREF expression levels. In this context it would be of interest to provide a further dissection of the WGS and KOCAL cohorts in relation to MYCN, MYC versus ALYREF expression levels. Also, it is not clear what the exact distribution is of low versus high risk cases in the WGS cohort as well as high risk MYCN amplified versus non-amplified. As 17q gain is also a very common genomic feature in high risk MYCN non-amplified cases further analysis of the possible functional role of ALYREF in context of MYCN or MYC levels could be of interest (based on the available expression data from the cohorts).
2. In fig2 NBLS is included in the low MYC NB group. The NBLS cell line has a structural rearrangement near the MYCN promotor presumably causing upregulation and has MYCN mRNA levels which are at the low end of the MYCN amplified cases but the highest expression for the MYCN non-amplified group. From the blots no MYC protein expression is visible. Also, for SK-N-FI 1 cannot see a signal for MYC protein expression.
3. In Fig. H versus J there seems to be a strong difference between effects of siRNA-1 knock down effects for MYCN mRNA versus protein expression.
4. With regard to the deletion experiments for MYCN-ALYREF interaction, can interaction with region A123-280 be excluded?
5. On line 474 the authors suggested the term oncogene for ALYREF, while this might be a semantic discussion, I would prefer to reserve this for genes with proven independent power to drive tumor formation (as for MYCN and LIN28B in the context of neuroblastoma) and therefore suggest to use the terms co-driver or cooperative oncogene.

Reviewer #2 (Remarks to the Author):

In the manuscript "An ALYREF-MYCN coactivator complex drives neuroblastoma tumorigenesis through effects on USP3 and MYCN stability", Marshall and colleagues identify the ALYREF protein as a factor that enhances MYCN protein stability and promotes neuroblastoma development. The authors show that ALYREF interacts with and promotes stabilization of MYCN. Mechanistically, the authors propose that ALYREF functions as a transcriptional co-activator of MYCN with the USP3 deubiquitinase being the critical target that mediates MYCN stabilization.

The findings are novel and of potential interest to the broad readership of the journal; the data are largely of high quality. However I have several concerns, both with methods used and data presented. I believe that addressing the following points will help clarify and strengthen the manuscript.

1. Cell viability assay (Figure 1l, etc) - what does this assay exactly measure and how does this relate to MYCN function - do the cell lines used have high rates of spontaneous apoptosis, which are reduced by MYCN in an ALYREF-dependent manner?

The assay used to determine cell proliferation (Figure 1m, etc) actually measures BrdU incorporation (= DNA synthesis) and not cell proliferation per se. Cumulative cell counting or colony formation are simpler assays that measure cell proliferation and should be used to support the authors' conclusions.

2. The motifs CACCTG and CAGCTG, identified within the MYCN binding sites at the ALYREF locus, are called canonical in the maintext (line 207) and non-canonical in the methods section (line 770). I think these are non-canonical E-Boxes and this should be made consistent in the manuscript.

Based on the provided ChIP-seq tracks (Figure 2f) these motifs appear not to lie within the MYCN peaks, at least for Kelly and NGP cell lines. Since MYCN binding to the ALYREF promoter is obvious from the published sequencing data, I suggest that authors test their ChIP experiments with primers that amplify the published MYCN peak summits, not adjacent candidate E-boxes.

3. How are the immunoprecipitation experiments that test MYCN ubiquitination performed (Figures 3e and 7c)? This assay should be done under denaturing conditions, otherwise it is impossible to argue that the observed ubiquitin chains are on MYCN - they can be conjugated to any MYCN-associated protein.

4. The authors use a very specific strategy based on several selection criteria to identify ALYREF target genes (page 12, Figure 5). However, it is not clear how it helps to identify true ALYREF targets and support the conclusion that ALYREF is a cofactor for MYCN. I don't understand why such criteria include selecting undefined "genes undergoing transcription" (line 294), and correlation with MYCN expression in tumors (this certainly biases towards MYCN-driven genes).

Instead, to show that ALYREF works as a (direct) co-factor for MYCN, the authors need to examine the binding sites for ALYREF from their ChIP-seq data for overlap with published MYCN sites and determine the effect of ALYREF shRNA on gene expression in cells with high and low MYCN levels.

5. The authors argue that USP3 is a target gene of MYCN/ALYREF which mediates the effects of ALYREF on MYCN stability. They use transient transfection to overexpress USP3 and rescue the effects of ALYREF knockdown (Figure 7). Since transient transfection experiments are prone to artefacts, this experiment should be performed with stable expression of USP3 (e.g., via lentiviral transduction).

6. Experiments using an shRNA against USP3 or a catalytically inactive USP3 variant are essential to validate USP3 as a key target of the ALYREF/MYCN complex in regulating MYCN stability and MYCN-dependent tumorigenesis.

7. The experiments showing co-recruitment of MYCN to the ALYREF binding site within the USP3 gene are not convincing (Figure 5e). First, they contradict published data for MYCN/RNAPII (Figure 5c) that show exclusive binding at the promoter, whereas ALYREF binds at a distal site around 6kb downstream. Second, the direct binding to the same genomic site may not be required for ALYREF to function as a cofactor for MYCN. The authors could examine available ChIP data for other transcription factors, chromatin conformation data, etc.

Also, the authors could consider the possibility that ALYREF stabilizes MYCN via direct binding and steric effects (e.g., by interfering with a recruitment of a Ub ligase) rather than downstream transcriptional regulation - the ALYREF-MYCN binding data are strong, in contrast to the data for co-recruitment of ALYREF and MYCN to the USP3 locus.

8. The authors should proofread the manuscript and rephrase sentences which are grammatically incorrect or scientifically inaccurate. For example, line 337: "USP3 expression associates with MYCN-amplification and is selectively dependent for neuroblastoma cell viability". Line 417: "Transcriptional regulators undergo rapid and timely degradation by ubiquitination".

Manuscript Number: NCOMMS-20-19257

Manuscript Title: An ALYREF-MYCN coactivator complex drives neuroblastoma tumorigenesis through effects on USP3 and MYCN stability

We would like to thank the editorial board and the reviewers for their comments and suggestions. The manuscript has been extensively revised, several new experiments and analyses have been conducted with new data added to Figures 1-2 and 5-7. A point by point response to the reviewer's comments and detailed explanation of all changes have been prepared as per following.

Key for responses

Reviewer Comments to the Author– Italics

Author Responses – Normal black text

Additions to the manuscript text are written in blue text

Changes in figures and supplementary figures are detailed first, followed by changes in figure legends, supplementary figure legends, results and materials and methods sections.

Reviewer #1

***Q1:** It is of interest that next to MYCN high level expression, also MYC impacts on ALYREF expression levels. In this context it would be of interest to provide a further dissection of the WGS and KOCAK cohorts in relation to MYCN, MYC versus ALYREF expression levels. Also, it is not clear what the exact distribution is of low versus high risk cases in the WGS cohort as well as high risk MYCN amplified versus non-amplified. As 17q gain is also a very common genomic feature in high risk MYCN non-amplified cases further analysis of the possible functional role of ALYREF in context of MYCN or MYC levels could be of interest (based on the available expression data from the cohorts).*

Response: We have performed additional analyses in multiple publicly available Neuroblastoma microarray/RNA-seq datasets with clinical annotations to investigate the correlation between MYC and ALYREF expression. We performed correlation analyses for ALYREF vs MYC expression, when considering all patients (regardless of risk stratification or MYCN-amplification status) in TARGET (WGS; n=154 and 249), Kocak (n=476), SEQC (n=498) and Versteeg (n=88) Neuroblastoma microarray/RNA-seq datasets. Based on our data, ALYREF mRNA expression does not significantly correlate with MYC mRNA expression (highest Pearson R=0.1343 from Kocak dataset; Rebuttal Figure 1) in any of the neuroblastoma patient tumour datasets.

As we do not have access to risk stratifications, investigating the distribution of low- versus high-risk cases and high-risk MYCN-amplified versus high-risk MYCN-non-amplified cases in the cohorts unfortunately was not possible. However, we were able to perform expression analysis of ALYREF based on MYCN-amplification status as clinical variable in the TARGET²⁵ (referenced in main manuscript; WGS; n=154 and 249) cohort (Rebuttal Figure 2) that is largely composed of Stage 4 high-risk neuroblastoma patients. Based on our analysis, ALYREF

expression was not significantly different between *MYCN*-amplified (n=120 and n=175) and *MYCN*-non-amplified (n=33 and n=68) cohorts.

We have also performed further (1) correlation analyses for *ALYREF* vs *MYC* and *ALYREF* vs *MYCN* expression levels, when considering only *MYCN*-amplified as well as only *MYCN*-non-amplified patients in the TARGET (WGS; n=154 and 249) and Kocak (n=476) cohort (Rebuttal Figure 3) and (2) Kaplan-Meier analysis when patients are sub-grouped by *MYCN*-amplification status (amp vs non-amp) and then *ALYREF* expression (median, upper quartile, upper decile) in the TARGET (WGS; n=154 and 249) cohort (Rebuttal Figure 4). *ALYREF* mRNA expression showed a slightly stronger correlation with *MYCN* expression when compared to *MYC* expression in both *MYCN*-amplified as well as *MYCN*-non-amplified patients in the TARGET (WGS; n=154 and 249) cohort, however the r values showed no significant linear correlations (Rebuttal Figure 3). *ALYREF* mRNA expression also showed stronger correlation with *MYCN* expression in *MYCN*-amplified patients when compared to *MYCN*-non-amplified patients in the Kocak (n=476) cohort, however, again, the r values were still considerably low (Rebuttal Figure 3). Kaplan Meier analyses also showed that *ALYREF* did not significantly affect overall or event-free patient survival in the TARGET (WGS; n=154 and 249) cohort.

Please note, we assessed and included all probes (for microarray) or RNA isoforms (RNA-seq) for *ALYREF* (also known as *THOC4*).

Based on the available expression data from the cohorts, we couldn't make any convincing conclusions regarding the link between *MYC* and *ALYREF* expression or regarding the possible functional role of *ALYREF* in high-risk *MYCN*-non-amplified cases in the current manuscript. No modifications have been made to the current manuscript following analyses.

Rebuttal Figure 1

a

b

c

d

Rebuttal Figure 1

Scatter plots, with a linear regression fit for neuroblastoma patients from the (a) TARGET (n=154 and 249), (b) Kocak (n=476), (c) SEQC (n=498) and (d) Versteeg (n=88) cohorts for *MYC* vs *ALYREF* gene expression (RNA-Seq, log₂RPM; Microarray, log₂). Patients with *MYCN*-amplifications (red), no amplifications (black) or are of unknown (grey) status are annotated. Pearson correlations as indicated on individual plots.

Rebuttal Figure 2

Rebuttal Figure 2

Boxplots displaying *ALYREF* mRNA expression (RNA-Seq, \log_2 RPM; Microarray, \log_2) based on *MYCN*-amplification status in the TARGET (WGS; n=154 and 249) neuroblastoma patient cohort. Adjusted p-value is provided from the transcriptome-wide differential gene expression analysis.

Rebuttal Figure 3

a

TARGET_RNASeq_MYCN-amplified only

TARGET_Microarray_MYCN-amplified only

b

TARGET_RNASeq_MYCN-non-amplified only

TARGET_Microarray_MYCN-non-amplified only

C

Kocak_Microarray_MYCN-amplified only

d

Kocak_Microarray_MYCN-non-amplified only

Rebuttal Figure 3

Scatter plots, with a linear regression fit for neuroblastoma patients from the TARGET (WGS; n=154 and 249) (a-b) and Kocak (n=476) (c-d) cohorts for *MYC* vs *ALYREF* and *MYCN* vs *ALYREF* gene expression (RNA-Seq, log₂RPM; Microarray, log₂) in *MYCN*-amplified (a and c) and *MYCN* non amplified (b and d) patient cohorts. Patients with *MYCN*- amplifications (red) and no amplifications (black) are annotated. Pearson correlations are as indicated on individual plots.

Rebuttal Figure 4

a

TARGET_RNASeq_EFS

TARGET_Microarray_EFS

b

TARGET_RNASeq_OS

TARGET_Microarray_OS

Rebuttal Figure 4

Kaplan-Meier survival curves showing the (a) event-free and (b) overall survival probability of patients in the TARGET cohort (WGS; n=154) when dichotomized by median, upper quartile and upper decile *ALYREF* gene expression. Hazard Ratio's (HR) and log-rank p-values are presented from a univariate CoxPH model.

Q2: In fig2 NBL5 is included in the low MYC NB group. The NBL5 cell line has a structural rearrangement near the MYCN promotor presumably causing upregulation and has MYCN mRNA levels which are at the low end of the MYCN amplified cases but the highest expression for the MYCN non-amplified group. From the blots no MYC protein expression is visible. Also, for SK-N-FI I cannot see a signal for MYC protein expression.

Response: The reviewer is correct, NBL5 cells have upregulated MYCN expression. We corrected the figure labelling and now refer to this cell line as a MYCN overexpressing cell line in the manuscript. For the ALYREF expression analysis (Figure 2c), we included the NBL5 cell line in the MYCN-amplified group. Unfortunately, due to the low MYCN expression in this cell line -when compared to other MYCN-amplified cells- we can only get a visible MYCN band in NBL5 cells when overexposing the membrane (Rebuttal Figure 5). We corrected the figure and added a MYCN blot with higher exposure time (but not overexposed) to see a clearer MYCN band in NBL5 cells (lane 7).

Rebuttal Figure 5

Rebuttal Figure 5

Representative western blot image of MYCN-amplified neuroblastoma cell lines for MYCN expression.

We mislabelled Figure 2b and accidentally swapped the labelling for “C-MYC overexpressing” versus “Low MYC expressing cells”. SK-N-FI and SH-EP cells have low MYC expression and it is now clearly indicated in New Figure 2b. Analysis on Figure 2c (including new p values) has been changed accordingly. Figure legend and result sections have been modified accordingly.

Changes in Figure:

Old Figure 2b-c

New Figure 2b-c

Changes in text:

Figure Legend

Page 45

...(b) Representative western blot and (c) densitometry analysis for ALYREF expression in a range of *MYCN*-amplified (SK-N-BE(2)C, Kelly, CHP-134, SK-N-DZ, LAN-1, IMR-32), *MYCN* overexpressing (NBLS), low MYC or *MYCN* expressing (SH-EP, SK-N-FI), cMYC overexpressing neuroblastoma (SH-SY5Y, SK-N-AS) and normal lung fibroblast (WI-38, MRC5) cells.

Results

Page 8-9

...Kocak³⁴ (n=476; Supplementary Fig. 2a) neuroblastoma patient tumor datasets. ALYREF protein expression levels were high in *MYCN*-amplified and *MYCN* overexpressing human neuroblastoma cells (lanes 1-7) across a panel of neuroblastoma cell lines (Fig. 2b-c), with significantly higher expression levels when compared to human fibroblasts (lanes 12 and 13; WI-38 and MRC5; Fig. 2b-c). Interestingly, two *MYCN*-non-amplified cell lines, with high cMYC expression ...

Q3. In Fig. H versus J there seems to be a strong difference between effects of siRNA-1 knock down effects for MYCN mRNA versus protein expression.

Response: The authors believe the reviewer refers to the expression difference in MYCN expression levels in Kelly cells (Figure 2i and Figure 2j) following MYCN siRNA-1 knock-down. Western blot experiments with all three experimental repeats at 48 hrs for MYCN knock-down using MYCN siRNA-1 or MYCN siRNA-2 (Rebuttal Figure 6) have been rerun to assess MYCN expression levels. In all three replicates, MYCN expression levels significantly decreased with no visible difference between MYCN siRNA-1 and MYCN siRNA-2. A new western blot image has been added to Figure 2i. New image panels now show efficient and consistent MYCN knock-down both on protein and mRNA levels in Kelly cells. There has been no changes to the text.

Rebuttal Figure 6

Rebuttal Figure 6

Representative western blot image of three biological replicates for transfection of MYCN-amplified Kelly cells with MYCN siRNA-1 or MYCN siRNA-2 at 48 hours.

Changes in Figure:

Old panel 2i

New panel 2i

Q4. With regard to the deletion experiments for MYCN-ALYREF interaction, can interaction with region A123-280 be excluded?

Response: The MYCN deletion mutant constructs used in our study were designed and kindly provided by Professor Wei Gu and colleagues (Institute for Cancer Genetics, Columbia University, New York, New York, USA, Herbert Irving Comprehensive Cancer Centre, Columbia University, New York, New York, USA). While the authors agree with the reviewer that adding this deletion mutant would further support the specificity of interaction between ALYREF and MYCN at AA280-342, deletion mutant AA 123-280 was not part of their original experimental design.

Q5: On line 474 the authors suggested the term oncogene for ALRYREF, while this might be a semantic discussion, I would prefer to reserve this for genes with proven independent power to drive tumor formation (as for MYCN and LIN28B in the context of neuroblastoma) and therefore suggest to use the terms co-driver or cooperative oncogene.

The authors agree, the requested change in the discussion has been made.

Changes in Text:

Page 21

Discussion

...Furthermore, ALYREF knockdown caused a significant decrease in cancer cell migration²³. Our *in vitro* and *in vivo* data strongly supports the role of ALYREF as a co-driver for oncogenesis in neuroblastoma. We also suggest that the regulatory relationship ...

Reviewer #2 (Remarks to the Author):

Q1a: Cell viability assay (Figure 11, etc) - what does this assay exactly measure and how does this relate to MYCN function - do the cell lines used have high rates of spontaneous apoptosis, which are reduced by MYCN in an ALYREF-dependent manner?

Response: Alamar Blue is an indicator of cell viability that detects metabolically active cells and has been applied to various aspects of monitoring cellular death, cell cycle function and compound toxicology. As the MYCN protein is situated downstream of several signalling pathways, it is involved in the control of these fundamental processes via promoting cell growth, proliferation, apoptosis and metabolism in different developing organs and tissues.

Alamar Blue monitors the reducing environment of the living cell. The active ingredient is resazurin (IUPAC name: 7-hydroxy-10-oxidophenoxazin-10-ium-3-one), which permeates through cell membranes. It is a blue non-fluorescent dye that is reduced to the pink-coloured, highly fluorescent resorufin upon entering cells. Resazurin acts as an intermediate electron acceptor in the electron transport chain and can be reduced by NADPH, FADH, FMNH, NADH, as well as the cytochromes in the living cells. As Resazurin accepts electrons, it changes from the oxidized, non-fluorescent, blue state to the reduced, fluorescent, pink state. This colour change can be detected using absorbance (as used in the present work, detected at 570 and 600 nm). Viable cells continuously convert resazurin to resorufin, thereby generating a quantitative measure of viability. As metabolic assay Alamar Blue may not accurately reflect cellular proliferation rates due to a non-linear correlation between dye reduction and cell number. In addition to mitochondrial reductases, other enzymes located in the cytoplasm and the mitochondria, such as dihydrolipoamine dehydrogenase, NAD(P)H:quinone oxidoreductase and flavin reductase may be able to reduce Alamar Blue. Hence, Alamar Blue reduction may signify an impairment of cellular metabolism and is not necessarily specific to interruption of electron transport and mitochondrial dysfunction.

We performed additional experiments and complemented our data obtained using Alamar Blue assay with BrdU assay and colony formation assays (please also refer to Q1b). These additional assays showed that ALYREF knock-down affected cell proliferation by the same magnitude as cell viability measured by Alamar Blue, implying that ALYREF mainly regulates cell proliferation and has little effect on cell death mechanisms. In addition, we also investigated whether ALYREF only causes a proliferation block in MYCN overexpressing cells or assists MYCN in regulating survival signals. Due to the positive forward feedback expression loop between MYCN and ALYREF proteins, investigation of the changes in apoptosis in MYCN-amplified neuroblastoma cell lines in an ALYREF-dependent manner is not possible. Therefore, to investigate the functional relationship between MYCN and ALYREF in regulating cell survival, we evaluated the phenotypic effect of ALYREF knock-down on human SHEPMYCN3 neuroblastoma cells which permit doxycycline-inducible overexpression of stably incorporated MYCN, therefore ideal for gene dependency experiments. SHEPMYCN3 cells transfected with control siRNA, ALYREF siRNA-1, or ALYREF siRNA-2 with or without doxycycline followed by multiple time point measurements of mitochondrial membrane potential (one of the earliest markers of apoptosis) using a MitoProbe DILC1(5) Assay and accumulation of SYTOX Green necrosis dye. Carbonyl cyanide 3-chlorophenylhydrazone (CCCP) was used as a positive control. Doxycycline and siRNA transfections at the time points used did not affect cell number, therefore were ideal conditions to conduct the measurements. We found no direct effect of ALYREF knock-down on cell death. The new cell survival data has been added to Supplementary Figure 1.

Supplementary figure legends, results and materials and methods sections (including new method description) have been modified accordingly.

Changes in Figures:

New Supplementary Figure 1j

New Supplementary Figure 1k

Changes in text:

Supplementary Figure Legend:

... colony formation were compared to the vehicle treated control siRNA (Vehicle siControl). Doxycycline induced (DOX) vs non-induced (Vehicle) SHEPMYCN3 neuroblastoma cells transfected with ALYREF siRNA-1, ALYREF siRNA-2 or Control siRNA (siControl) subjected to (j) MitoProbe DilC1(5) and (k) SYTOX Green measurements at 16-48 hours. CCCP served as positive control. Differences in fluorescence signals were normalized to the vehicle treated control siRNA (Vehicle siControl). Respective p values (p) are displayed. Multiple comparisons were not significant unless otherwise noted. (l) Representative immunoblots of ALYREF expression ...

Results:

Page 7-8

...Taken together these data showed that MYCN-induced increases in neuroblastoma cell viability and proliferation were, in part, dependent on ALYREF. ALYREF is known to regulate cell proliferation^{23,24}, however its role in regulating cell survival is unknown. To determine whether ALYREF knock-down affects MYCN-induced increases in cell survival, we investigated the effect of ALYREF inhibition on cell death in doxycycline-induced (DOX) or not induced (Vehicle) SHEPMYCN3 cells, transfected with control siRNA, ALYREF siRNA-1, or ALYREF siRNA-2 (Supplementary Fig. 1j-k). Abolition of mitochondrial membrane potential is one of the earliest markers of apoptosis. Mitochondrial membrane potential was measured using a MitoProbe DilC1(5) Assay (Supplementary Fig. 1j). MYCN overexpression induced by doxycycline treatment did not affect mitochondrial membrane potential (Supplementary Fig. 1j) at the earlier time points. Only at 48 hours was there a slight decrease in mitochondrial membrane potential induced by MYCN overexpression. ALYREF knock-down did not significantly affect the survival of the doxycycline-induced (DOX) or not induced (Vehicle) SHEPMYCN3 cells. To further investigate necrotic processes consequent on altered ALYREF levels, we assessed the effects of ALYREF knock-down on MYCN-induced survival by measuring SYTOX Green accumulation in the cells (Supplementary Fig. 1k). The dye is able to penetrate (and then bind to the nucleic acids) only to necrotic cells with ruptured plasma membranes, whereas healthy cells with intact surface membranes show significantly lower SYTOX Green staining. As expected, doxycycline-induced MYCN overexpression protected the cells from cell death and significantly decreased the number of necrotic SHEPMYCN3 cells (Supplementary Fig. 1k). However, ALYREF knock-down did not further decrease SYTOX Green accumulation in the cells. Therefore, we conclude that ALYREF predominantly causes induction of proliferation in MYCN overexpressing cells, but does not assist MYCN in regulating cell survival. We also investigated the phenotypic effect of ALYREF knock-down on...

Determination of cell death

SHEPMYCN3 neuroblastoma cells (5×10^3 cells per well) were seeded in 96-well plates, transfected with control siRNA, ALYREF siRNA-1, or ALYREF siRNA-2 for 16-48 hours with and without doxycycline. Mitochondrial membrane potential of cells was determined using a MitoProbe DiI C1(5) Assay Kit (Life Technologies). The cells were incubated with DiI C1(5) working solution (50 nM/well) for 30 min, and the fluorescence of DiI C1(5) was measured at 630-nm excitation and 670-nm emission wavelengths using VICTOR Multilabel Counter (Perkin Elmer). Dead cells were determined by SYTOX Green staining (Life Technologies). The cells were incubated with SYTOX Green working solution (30 nM/well) for 30 min, and the fluorescence of SYTOX Green was measured at 490-nm excitation and 520-nm emission wavelengths using VICTOR Multilabel counter (Perkin Elmer).

***Q1b:** The assay used to determine cell proliferation (Figure 1m, etc) actually measures BrdU incorporation (= DNA synthesis) and not cell proliferation per se. Cumulative cell counting or colony formation are simpler assays that measure cell proliferation and should be used to support the authors' conclusions.*

Response: We performed additional colony formation assays (for both newly acquired data and results detailed in the original manuscript) to support our conclusions regarding the changes in cell proliferation measured by BrdU assays. As we found it difficult to accurately count colonies in cell lines with small and less dense colonies (e.g. SHEPMYCN3 cells), we measured the absorbance of crystal violet instead of using colony number for quantification. The new as well as modified data including colony formation assay images and quantification have been added to the main Figures and Supplementary Figures. Figure legends, Results and Methods sections have been modified accordingly.

Changes in Figures:

New Figure 1n

New Supplementary Figure 1i

New Figure 7d

New Supplementary Figure 7a

Modified Figure 8c

Old Figure 8C

New Figure 8C

Uncropped and unprocessed colony formation assay images of the representative replicate used in main figures have been added to the Supplementary File, related to figures as indicated.

Fig.1n

Fig.7d

Fig.8c

Changes in text

Materials and Methods:

Page 34:

Colony formation

200 SHEPMYCN3 cells transfected with control siRNA, ALYREF siRNA-1, or ALYREF siRNA-2 were seeded in 6-cm² plates and treated with and without doxycycline (Sigma) for 12 days. 250 SK-N-BE(2)C and 500 Kelly cells expressing control siRNA, USP3 siRNA-1, or USP3 siRNA-2 were seeded in 6-cm² plates and grown for 10 and 12 days respectively. 250 SK-N-BE(2)C and 500 Kelly cells expressing shALYREF or control shControl constructs were seeded in 6-cm² plates and kept under doxycycline treatment for 10 and 12 days respectively, to allow colonies to form. Colonies were fixed and stained with crystal violet solution (Sigma) then washed with water to remove unincorporated stain. Cells were photographed and colony formation was quantified from crystal violet absorbance readings at 590 nm on the Benchmark Plus microplate reader (Bio-Rad).

Figure Legends:

Page 45, related to New Figure 1n

...to the vehicle (Vehicle) treated siControl. (n) Doxycycline induced (DOX) vs non-induced (Vehicle) SHEPMYCN3 neuroblastoma cells transfected with ALYREF siRNA-1, ALYREF siRNA-2 or Control siRNA (siControl) were grown for 12 days, followed by colony formation assay. All *in vitro* experiments were repeated ...

Page 52, related to New Figure 7d

...were compared to Control siRNA expressing cells. (d) Kelly and SK-N-BE(2)C cells transfected with USP3 siRNA-1, USP3 siRNA-2 or Control siRNA were grown for 12 days (Kelly) or 10 days (SK-N-BE(2)C), followed by colony formation assay. (e) Immunoblot analysis of USP3 expression ...

Supplementary File

related to New Figure 1n

... are ranked by median copy number. (i) Quantification of colony forming assay for doxycycline induced (DOX) vs non-induced (Vehicle) SHEPMYCN3 neuroblastoma cells transfected with ALYREF siRNA-1, ALYREF siRNA-2 or Control siRNA (siControl) based on crystal violet absorbance (590nm). Differences in colony formation were compared to the vehicle treated control siRNA (Vehicle siControl). Doxycycline induced (DOX) vs non-induced (Vehicle) SHEPMYCN3 neuroblastoma cells ...

related to New Figure 7d

(a) Quantification of colony forming assay for Kelly and SK-N-BE(2)C neuroblastoma cells transfected with USP3 siRNA-1, USP3 siRNA-2 or Control siRNA (siControl) based on crystal violet absorbance (590nm). Differences in colony formation were compared to the control siRNA (siControl) expressing cells. (b) Stable Kelly and SK-N-BE(2)C cells overexpressing USP3 (USP3) ...

Results:

Page 7, related to New Figure 1n and New Supplementary Fig 1i

“MYCN increases neuroblastoma cell viability in an ALYREF-dependent manner”

...ALYREF siRNA-1, or ALYREF siRNA-2 (Fig. 1k) for 72 hours with (lanes 4-6) and without doxycycline (lanes 1-3). As expected, transfection of ALYREF siRNA in MYCN-low and ALYREF-low expressing conditions did not reduce cell viability (Fig. 1l) or proliferation (Fig. 1m-n and Supplementary Fig. 1i). However, MYCN-induced increases in viability (Fig. 1l) and proliferation (Fig. 1m-n) and Supplementary Fig. 1i) were blocked by ALYREF knock-down. Taken together these data showed that MYCN-induced increases in neuroblastoma cell viability and proliferation were, in part, dependent on ALYREF.

Page 17, related to New Figure 7d and New Supplementary Figure 7a

... Alamar blue assays showed that Kelly and SK-N-BE(2)C cells transfected with USP3 siRNA-1 or USP3 siRNA-2 displayed reduced cell viability (Fig. 7b) when compared to Control siRNA expressing cells. In addition, knocking down USP3 decreased both short-term (Fig. 7c) and long-term (Fig. 7d and Supplementary Fig. 7a) cell proliferation of Kelly and SK-N-BE(2)C cells. To investigate whether USP3 is responsible ...

Q2a: The motifs CACCTG and CAGCTG, identified within the MYCN binding sites at the ALYREF locus, are called canonical in the maintext (line 207) and non-canonical in the methods section (line 770). I think these are non-canonical E-Boxes and this should be made consistent in the manuscript.

Response: The reviewer is correct, both motifs CACCTG and CAGCTG are non-canonical. Results and Figure Legend sections have been revised and made consistent.

Changes in text:

Results

Page 9

...correlated with *MYCN* expression in ganglia tissues throughout tumor progression (Fig. 2e). Analysis of the *ALYREF* gene promoter sequence and published chromatin-immunoprecipitation (ChIP)-sequencing data³⁶ revealed non-canonical *MYCN* binding sites CACCTG (at -549 bp) and CAGCTG (+ 635 bp) in close proximity to the *ALYREF* transcription start site (TSS, Fig. 2f).

Figure Legends

Page 46

(f) Schematic representation of the *ALYREF* gene promoter in a panel of *MYCN*-amplified neuroblastoma cells detailing canonical (CACGCTG) and non-canonical (CANNTG) *MYCN* binding motifs.

Q2b: Based on the provided ChIP-seq tracks (Figure 2f) these motifs appear not to lie within the MYCN peaks, at least for Kelly and NGP cell lines. Since MYCN binding to the ALYREF promoter is obvious from the published sequencing data, I suggest that authors test their ChIP experiments with primers that amplify the published MYCN peak summits, not adjacent candidate E-boxes.

Response: We designed a new set of primers, performed additional experiments and revised the ChIP experiments to support our claim that *ALYREF* is a transcriptional target of *MYCN*. The new data has been added to Figure 2g-h. Figure legend, results and materials and methods sections have been modified accordingly including the new primer sequences.

Changes in Figures:

Old Figure 2g and 2h

New Figure 2g and 2h

Changes in Text:

Figure Legend:

Page 46

...and non-canonical (CANNTG) MYCN binding motifs. (g) Schematic representation of the ALYREF target sequence (Promoter) and negative control (Control) target sequence used for RT-PCR following chromatin-immunoprecipitation (ChIP), detailing the MYCN peak summit and its distance from transcription start site (TSS). (h) ChIP assays were performed with a control IgG (IgG) or anti-MYCN antibody (MYCN), followed by PCR with primers targeting the negative control region (Control; 2000 bp downstream of TSS) or the ALYREF gene promoter containing the MYCN peak summit (Promoter, -248 bp downstream of TSS) in Kelly and SK-N-BE(2)C cells. Fold enrichment of the ALYREF gene promoter...

Results:

Page 9-10

...correlated with MYCN expression in ganglia tissues throughout tumor progression (Fig. 2e). Analysis of the ALYREF gene promoter sequence and published chromatin-immunoprecipitation (ChIP)-sequencing data³⁶ revealed non-canonical MYCN binding sites CACCTG (at -549 bp) and CAGCTG (+635 bp) in close proximity to the ALYREF transcription start site (TSS, Fig. 2f). We performed ChIP assays with an anti-MYCN antibody or control IgG antibody followed by real-time PCR with primers targeting a

negative control (Control; 2000 bp upstream TSS) and the MYCN peak summit (Promoter, -248 bp) upstream of the ALYREF TSS (Fig. 2g). ChIP assays showed significant MYCN binding^{14,37,38} at its putative binding site 248 bp upstream ALYREF TSS. The MYCN antibody immunoprecipitated the ALYREF region 3-fold higher than the negative control region in SK-N-BE(2)C cells and 6-fold in Kelly cells (Fig. 2h), confirming that MYCN directly binds the ALYREF gene promoter. Transfection of MYCN-amplified SK-N-BE(2)C, Kelly (Fig. 2i-j) and CHP-134 cells ...

Materials and Methods:

Page 28

Primer sequences (5'→3') for chromatin immunoprecipitation: NEGATIVE CONTROL forward -ACATGGGTACCAACCACCTG-, reverse -AGGCTGGTCTCGAACTGTCA-; ALYREF forward -GTAGGGCGGTGCGTGATTAG-, reverse -TAGGCTCCGCCTCCAACG-; USP3 forward -CACTGTTTCAGACTTTTAGCTTGC-, reverse -GGGGGTAACAAAAGACCAAATTTTA-.

Q3: *How are the immunoprecipitation experiments that test MYCN ubiquitination performed (Figures 3e and 7c)? This assay should be done under denaturing conditions, otherwise it is impossible to argue that the observed ubiquitin chains are on MYCN - they can be conjugated to any MYCN-associated protein.*

Response: Post MG-132 treatment, cells were lysed with denaturing lysis buffer (6 M guanidine-HCl, 50 mM sodium phosphate, pH 8.0, 500 mM NaCl, 5 mM imidazole) and cleared by centrifugation at 16,000 RCF for 20 min. 5% of protein was saved to serve as an input. The rest of the sample (300-750 ug total protein) was diluted in BC100 buffer freshly supplemented with protease inhibitor cocktail (Sigma-Aldrich) and subjected to immunoprecipitation with either 5 ug MYCN-specific antibody (Merck Millipore), or with 5 ug control IgG antibody (Santa Cruz Biotechnology) and A/G PLUS agarose beads (Santa Cruz Biotechnology) at 4 °C overnight. Prior to immunoprecipitation, the beads were washed three times with denaturing wash buffer (8 M urea, 50 mM sodium phosphate, pH 8.0, 500 mM NaCl). After overnight incubation at 4 °C, beads were washed five times with cold BC100 buffer and once with cold sodium chloride solution (150 mM). Proteins were eluted in Laemmli sample buffer (62.5 mM Tris-HCl, pH 6.8; 25% glycerol; 2% SDS; 0.01% Bromophenol Blue) supplemented with β-mercaptoethanol at a final concentration of 5% (710 mM) at a 4 parts sample to 1 part sample buffer dilution, and separated by electrophoresis using 10.5% or 10-14% Tris-HCl Criterion gels (Bio-Rad, Gladesville, NSW, Australia).

Materials and methods section have been modified and now includes a more detailed description for ubiquitin experiments.

Changes in text:

Materials and Methods

Page 32

Ubiquitin assays

Cells were exposed to the proteasomal inhibitor MG132 (Sigma) at 20 μ M concentration⁴². Four hours post MG-132 treatment, cells were lysed with denaturing lysis buffer (6 M guanidine-HCl, 50 mM sodium phosphate, pH 8.0, 500 mM NaCl, 5 mM imidazole). 5% of protein was saved to serve as an input. The rest of the sample (750 μ g total protein) was diluted in BC100 buffer (Sigma) freshly supplemented with protease inhibitor cocktail (Sigma) and subjected to immunoprecipitation with either 5 μ g MYCN-specific antibody (Merck Millipore), or with 5 μ g control IgG antibody (Santa Cruz Biotechnology) and A/G PLUS agarose beads (Santa Cruz Biotechnology) at 4 °C overnight. Prior to immunoprecipitation, the beads were washed three times with denaturing wash buffer (8 M urea, 50 mM sodium phosphate, pH 8.0, 500 mM NaCl). After overnight incubation at 4 °C, beads were washed five times with cold BC100 buffer (Sigma) and once with cold sodium chloride solution (150 mM). Proteins were eluted in Laemmli sample buffer (62.5 mM Tris-HCl, pH 6.8; 25% glycerol; 2% SDS; 0.01% Bromophenol Blue) supplemented with β -mercaptoethanol (Bio-Rad) at a final concentration of 5%, and separated by electrophoresis using 10.5% or 10-14% Tris-HCl Criterion gels (Bio-Rad).

Q4: The authors use a very specific strategy based on several selection criteria to identify ALYREF target genes (page 12, Figure 5). However, it is not clear how it helps to identify true ALYREF targets and support the conclusion that ALYREF is a cofactor for MYCN. I don't understand why such criteria include selecting undefined "genes undergoing transcription" (line 294), and correlation with MYCN expression in tumors (this certainly biases towards MYCN-driven genes). Instead, to show that ALYREF works as a (direct) co-factor for MYCN, the authors need to examine the binding sites for ALYREF from their ChIP-seq data for overlap with published MYCN sites and determine the effect of ALYREF shRNA on gene expression in cells with high and low MYCN levels.

Response: As the current manuscript aims to describe the mechanism between 17q21ter gain/ALYREF and MYCN stability required for neuroblastoma tumorigenesis, we were looking for ALYREF target genes showing strong association with MYCN expression levels, neuroblastoma patient outcome and having known biological function relevant to ubiquitination, hence the stringent and specific selection criteria detailed in the paper. We did not aim to introduce ALYREF as a transcriptional co-factor for MYCN in the conventional sense (overlapping binding sites on the chromatin) but were rather looking for target genes/mechanisms that could highlight a previously unrecognised role of ALYREF, to promote long-range regulatory chromatin interactions that support MYCN-driven neuroblastoma phenotype. With all the new mechanistic experiments detailed under Question 7, we believe that we have now adequately addressed the mechanism. We do not wish to mislead the readers, therefore in the revised version of the manuscript we changed the terminology and do not use the term "direct co-factor for MYCN" when describing the regulatory role of ALYREF on the chromatin. We have also removed the criteria "undergoing active transcription" and replaced it with "presence of MYCN...at target gene promoters". We made changes to Results as well as Discussion sections.

Changes in text:

Results

Page 12-14

ALYREF has been suggested to act as a regulator of DNA binding and modulate the activity of transcription factors¹⁵⁻¹⁸. The binding of ALYREF and MYCN proteins in the nucleus as well as the positive forward feedback expression loop between the two proteins led us to hypothesize that ALYREF and MYCN cooperated and played a role in upregulating transcriptional programs involved in the ubiquitination process regulating MYCN turnover. To identify gene targets, we first mapped the genome-wide profile of ALYREF-chromatin interactions in SK-N-BE2C cells using ChIP-seq. We looked for transcriptional target genes of ALYREF based on the following selection criteria (Fig. 5a): (1) presence of MYCN and RNA Pol II (indicating active transcriptional elongation), as well as two proximate histone modifications associated with transcriptional activation (H3K27ac and H3K4me3) at the gene promoter; (2) having a high fold enrichment (FE)...

...in both Kelly and SK-N-BE(2)C cells, respectively (Fig. 5e). Binding of ALYREF to MYCN on chromatin might be due to a yet unknown function of ALYREF in regulating long-range interactions involving chromatin looping. To confirm the potential interaction between the USP3 promoter sequence binding MYCN at the TSS and the DNA sequence containing the ALYREF binding site at +6800 bp from the TSS, a chromatin conformation capture (3C) assay (Fig. 5f) was performed ...

Discussion

Page 20

As MYCN showed dependency on ALYREF for efficient induction of *USP3* expression and the two proteins interact via DNA looping, we argue that the binding of ALYREF to MYCN at AA 281-345, enhances the affinity of MYCN to the chromatin. Genome-wide intergenic binding of...

Nonetheless, we have performed further bioinformatics analyses -as per the reviewer's request- and examined the ALYREF binding sites for overlap with published MYCN sites. We identified 8 overlaps between ALYREF and MYCN peaks on the chromatin in SK-N-BE(2)C cells (Rebuttal Table 1 "peak_overlaps_ALYREF_MYCN_BE(2)C"). After conducting literature review, we were not able to identify connections between MYCN stability and target gene candidates identified from our new analyses. To investigate the role of further ALYREF target genes with less bias selection criteria is a natural extension to our study, however in the current manuscript we focus on describing the mechanism underlying the ALYREF-USP3-MYCN regulatory circuit.

Q5: The authors argue that USP3 is a target gene of MYCN/ALYREF which mediates the effects of ALYREF on MYCN stability. They use transient transfection to overexpress USP3 and rescue the effects of ALYREF knockdown (Figure 7). Since transient transfection experiments are prone to artefacts, this experiment should be performed with stable expression of USP3 (e.g., via lentiviral transduction).

Response: To investigate whether USP3 is responsible for the ALYREF-induced increase in MYCN stability and neuroblastoma cell growth, we created two pools of clones from each of the neuroblastoma cell lines Kelly and SK-N-BE(2)C cells (by lentiviral transduction) stably overexpressing either a vector control (Vector) or USP3 (USP3) ORF, and evaluated the effect of USP3 overexpression on MYCN ubiquitination and neuroblastoma cell viability following ALYREF knock-down. Both Alamar blue and ubiquitin assays showed that stable overexpression of USP3 reversed the effect of ALYREF knockdown on neuroblastoma cell viability and MYCN ubiquitination, confirming that USP3 is a target gene that mediates the effects of ALYREF on MYCN stability. Original figures and data (Old Figure 7) using transient USP3 overexpression have been removed from the manuscript, new data using stable USP3 overexpressing cells have been added as new main Figure 7 (New Figure 7) and Supplementary Figure 7. Figure legends, results and materials and methods sections (containing description of new techniques used specific to stable cell line generation) have been modified accordingly.

Changes in Figures:

Old Figure 7

New Figure 7 containing new data panels e-g

Changes in Supplementary Figures:
New Supplementary Figure 7b

Changes in text:

Figure Legends

Figure 7

Page 52

... followed by colony formation assay. (e) Immunoblot analysis of USP3 expression in stable Kelly and SK-N-BE(2)C cells overexpressing USP3 (USP3) or Vector control (Vector). NTC, non-transduced control cells. (f) Stable Kelly and SK-N-BE(2)C cells overexpressing USP3 (USP3) or Vector control (Vector) were co-transfected with ALYREF siRNA-1 or ALYREF siRNA-2 or Control siRNA, followed by cell viability measurements at 72 hours. Differences in cell viability were compared to Vector Control siRNA expressing cells. (g) Stable Kelly and SK-N-BE(2)C cells overexpressing USP3 (USP3) or Vector control (Vector) were co-transfected with ALYREF siRNA-1, or ALYREF siRNA-2 or control siRNA, followed by treatment with MG132 for 4 hours. Cells were then subjected to endogenous MYCN immunoprecipitation and immunoblot analyses for K-63 and K-48-linked ubiquitination. All *in vitro* experiments were repeated at least three times, except ubiquitin assay (n=2). Error bars, SEs. Respective p-values (p) are displayed. Comparisons were not significant unless otherwise noted.

Supplementary Figure Legends

Supplementary Figure 7

... control siRNA (siControl) expressing cells. (b) Stable Kelly and SK-N-BE(2)C cells overexpressing USP3 (USP3) or Vector control (Vector) were co-transfected with ALYREF siRNA-1 or ALYREF siRNA-2 or control siRNA, followed by treatment with MG132 for 4 hours. Cells were then subjected to endogenous MYCN immunoprecipitation and immunoblot analyses for K-63 and K-48-linked ubiquitination. All *in vitro* experiments were repeated at least three times, except rescue ubiquitination assay (n=2). Error bars, SEs. Respective p values (p) are displayed. Comparisons were not significant unless otherwise noted.

Results

Page 17

...To investigate whether USP3 is responsible for the ALYREF-induced increase in MYCN stability and neuroblastoma cell growth, we created pool of clones Kelly and SK-N-BE(2)C cells (Fig. 7e) stably overexpressing either Vector control (Vector; lane 2) or USP3 (USP3; lane 3), and evaluated the effect of USP3 overexpression on MYCN ubiquitination and neuroblastoma cell viability following ALYREF knock-down. Kelly and SK-N-BE(2)C cells stably overexpressing vector control (Vector) or USP3 were transfected with control siRNA, ALYREF siRNA-1 or ALYREF siRNA-2 (Fig. 7f-g). Alamar blue assays for cell viability showed that knocking down ALYREF reduced the number of viable Kelly and SK-N-BE(2)C cells, however, stable overexpression of USP3 reversed the effect of ALYREF knockdown (Fig. 7f). We next examined the hypothesis that the ALYREF effects on MYCN ubiquitination were USP3 dependent. As expected, knocking down ALYREF (lanes 2 and 3; Fig. 7g

“INPUT”) decreased USP3 and MYCN expression levels in both Kelly and SK-N-BE(2)C cells (lanes 2 and 3; Fig. 7g “INPUT”). USP3 overexpression however, significantly increased MYCN expression levels in both cell lines (lanes 5 and 6; Fig. 7g “INPUT”). We immunoprecipitated endogenous MYCN using a MYCN antibody (lanes 1-6; Fig. 7g “IP”) or control IgG antibody (lanes 7-12; Supplementary Fig. 7b “IP”) from vector control (Vector) or USP3-overexpressing cells that had been treated with siRNAs for ALYREF then probed with antibodies specific for K48- or K63-linked ubiquitin chains. The results showed that ALYREF knock-down increased K48- and K63-linked polyubiquitination of MYCN (lanes 1-3; Fig. 7g “IP”), whereas forced overexpression of USP3 partially reversed this change in ubiquitination (lanes 4-6; Fig. 7g “IP”). These findings suggest that USP3 is the effector protein for the effects of the MYCN-ALYREF-USP3 signal on neuroblastoma cell viability and MYCN stability.

Methods

Page 23

Generation of USP3 overexpressing neuroblastoma cell lines

To generate stable USP3 overexpressing neuroblastoma cells, SK-N-BE(2)C and Kelly cells were infected with Precision LentiORF encoding control ORF or Precision LentiORF encoding USP3 ORF lentiviral particles (OHS5899-202618161 and OHS5900-202620492) purchased from Dharmacon (Millenium Science Australia). The viral particles were incubated with polybrene (Santa Cruz Biotechnology) and added to SK-N-BE(2)C and Kelly cells. 7.5 ug/ml blasticidin (Sigma) was used to select positive clones.

Uncropped and unprocessed western blot images of the representative replicate used in main figures have been added to the Supplementary File, related to figures as indicated.

Fig.7e

Fig.7g

Q6: Experiments using an *shRNA* against *USP3* or a catalytically inactive *USP3* variant are essential to validate *USP3* as a key target of the *ALYREF/MYCN* complex in regulating *MYCN* stability and *MYCN*-dependent tumorigenesis.

Response: To validate *USP3* as a key target of the *ALYREF/MYCN* complex in regulating *MYCN* stability, we further examined how *USP3* expression affects *MYCN* protein levels and stability using wild-type (*USP3*^{wt}) and catalytically inactive *USP3* (*USP3*^{C168S})^{46,47,49,50} (referenced in main text) variant. We found that (1) *MYCN* protein levels increased after *USP3* overexpression in a dose-dependent manner, (2) the ectopically overexpressed *USP3*^{C168S} mutant had a significantly reduced effect on *MYCN* protein expression and half-life when compared to *USP3*^{wt}, and (3) the *USP3*^{C168S} mutant was unable to increase deubiquitination of *MYCN*. New data have been added to Figure 6 as new Figure 6h-l. Figure legends, Supplementary Figure Legends, Results and Methods sections have been modified accordingly.

To validate *USP3* as a key target of the *ALYREF/MYCN* complex in regulating *MYCN* - driven tumourigenesis, we silenced endogenous *USP3* expression in Kelly and SK-N-BE(2)C *MYCN*-amplified neuroblastoma cells using two different *USP3* siRNA sequences, as well as Control siRNA. *USP3* knock-down in both *MYCN*-amplified neuroblastoma cell lines significantly decreased *MYCN* protein expression and had marked effects on the cancer phenotype as shown by a marked reduction in cell viability, as well as short-term and long-term cell proliferation of Kelly and SK-N-BE(2)C cells. New data have been added to Figure 7 as new Figure 7a-d. Figure legends, Supplementary Figure Legends, Results and Methods sections have been modified accordingly.

Changes in Figures:

New Figure 6 panels

New Figure 7 panels

Changes in Supplementary Figures:

New Supplementary Figure 6 panels

New Supplementary Figure 7 panels

a

Changes in Text:

Figures Legend

Figure 6

Page 51

...the cell lysate was loaded for input. (h) Schematic representation of wild-type (USP3^{wt}) and mutant (USP3^{C168S}) USP3 proteins. ZnF, zinc finger ubiquitin binding domain (ZnF UBP); USP, ubiquitin-specific protease domain. All constructs were Flag-tagged. (i) Representative Western blot analysis for the indicated ectopically overexpressed wild-type (USP3^{wt}) and mutant (USP3^{C168S}) USP3 proteins as well as HA-MYCN from HEK293T cells. Vector indicates HEK293T cells that were transfected with an empty vectors. (j) Representative immunoblot analysis for the indicated ectopically overexpressed HA-MYCN and Flag-USP3^{wt} or Flag-USP3^{C168S} from HEK293T cells after immunoprecipitation of HA-MYCN. 5% of the cell lysate was loaded for input. Vector and “-” indicates HEK293T cells that were transfected with an empty pCMV3Tag3A or pCDNA3.1 vector, respectively. (k) HEK293T cells expressing wild-type (USP3^{wt}) and mutant (USP3^{C168S}) USP3 proteins were treated with 100 ug/ml cycloheximide (CHX) for 0 (no CHX), 30, 45, or 60 mins. Protein was extracted from the cells and subjected to immunoblot analysis of MYCN and Flag-USP3 expression. (l) HEK293T cells expressing HA-MYCN, His-Ubiquitin, wild-type (USP3^{wt}) and mutant (USP3^{C168S}) USP3 proteins or empty vector (Vector) were treated with MG132 for 4 hours. Cells were then subjected to HA-MYCN immunoprecipitation and immunoblot analyses for ubiquitination. All *in vitro* experiments were repeated at least three times, except ubiquitin assay (n=2). Error bars, SEs. Respective p-values (p) are displayed. Comparisons were not significant unless otherwise noted.

Supplementary Figure Legend

Supplementary Figure 6

...cell lines were compared using one-way ANOVA. (f) Representative immunoblot of HEK293T cells expressing HA-MYCN and either empty vector control (“-“) or increasing amount of Flag-USP3^{wt} at 24 hours. (g) Densitometry analysis of CHX assay. MYCN protein levels were normalized by actin, the ratio of MYCN protein/actin protein were artificially set as 1.0 for samples untreated with CHX to obtain half-life ($T_{1/2}$) of MYCN. (h) HEK293T cells expressing HA-MYCN, His-Ubiquitin, wild-type (USP3^{wt}) and mutant (USP3^{C168S}) USP3 proteins or empty vector (Vector) were treated with MG132 for 4 hours. Cells were then subjected to HA-MYCN immunoprecipitation and immunoblot analyses for ubiquitination. All *in vitro* experiments were repeated at least three times, except ubiquitin assay (n=2). Error bars, SEs. Respective p-values (p) are displayed. Comparisons were not significant unless otherwise noted.

Figures Legend

Figure 7

Page 51-52

(a) Immunoblot analysis of MYCN and USP3 expression in Kelly and SK-N-BE(2)C cells following siRNA-mediated USP3 knockdown for 48 hours. Cell viability (b) and proliferation (c) analysis of Kelly and SK-N-BE(2)C cells transfected with USP3 siRNA-1, USP3 siRNA-2 or Control siRNA for 96 hours. Differences in cell growth were compared to Control siRNA expressing cells. (d) Kelly and SK-N-BE(2)C cells transfected with USP3 siRNA-1, USP3 siRNA-2 or Control siRNA were grown for 12 days (Kelly) or 10 days (SK-N-BE(2)C), followed by colony formation assay. (e) Immunoblot analysis of USP3 expression in stable...

Supplementary Figure Legend

Supplementary Figure 7

(a) Quantification of colony forming assay for Kelly and SK-N-BE(2)C neuroblastoma cells transfected with USP3 siRNA-1, USP3 siRNA-2 or Control siRNA (siControl) based on crystal violet absorbance (590nm). Differences in colony formation were compared to the control siRNA (siControl) expressing cells.

Results

Figure 6

Page 16

... using the control IgG antibody (lanes 2 and 4; Fig. 6g). To understand the functional consequence of this interaction, we measured the effects of USP3 expression on MYCN protein levels. We found that MYCN protein levels increased after USP3 overexpression

(Supplementary Fig. 6f) in dose-dependent manner. Furthermore, ectopically expressed USP3 mutant (Fig. 6h) lacking the catalytic activity (USP3^{C168S})^{46,47,49,50} had a significantly reduced effect on MYCN protein expression (Fig. 6i), suggesting that USP3 regulates MYCN expression through its de-ubiquitinase activity. Although both ectopically expressed USP3^{wt} and USP3^{C168S} coimmunoprecipitated with MYCN (Fig. 6j), the half-life of MYCN was only significantly extended in the presence of USP3^{wt} (lanes 5-8) but not USP3^{C168S} (lanes 9-12) or an empty (lanes 1-4) expression vector (Fig. 6k and Supplementary Fig. 6g). Despite their interaction, the USP3^{C168S} mutant was unable to catalyze de-ubiquitination of MYCN (lane 5) in HEK293T cells (Fig. 6l), suggesting that the catalytic activity is required for USP3 to impact MYCN ubiquitination levels. Collectively, these results demonstrate that MYCN is a target of USP3.

Figure 7

Page 17

We further investigated the phenotypic effect of USP3 knock-down in *MYCN*-amplified neuroblastoma cell lines with high USP3 expression. Silencing endogenous USP3 with USP3 siRNA-1 or USP3 siRNA-2 (lanes 2-3 and 5-6, Fig. 7a) in *MYCN*-amplified neuroblastoma cells significantly decreased MYCN protein expression (Fig. 7a). Knockdown of USP3 had marked effects on the cancer phenotype in *MYCN*-amplified neuroblastoma cells. Alamar blue assays showed that Kelly and SK-N-BE(2)C cells transfected with USP3 siRNA-1 or USP3 siRNA-2 displayed reduced cell viability (Fig. 7b) when compared to Control siRNA expressing cells. In addition, knocking down USP3 decreased both short-term (Fig. 7c) and long-term (Fig. 7d and Supplementary Fig. 7a) cell proliferation of Kelly and SK-N-BE(2)C cells. To investigate whether USP3 is responsible for the ALYREF-induced increase in MYCN stability ...

Materials and Methods specific to Figure 6 and Figure 7 (Ubiquitin assays and colony formation assays as described above)

Page 26-27

siRNA and plasmid DNA transfections

...synthesized by Qiagen (VIC, Australia). ON-TARGETplus Human USP3 siRNA (LQ-006078-00-0020) synthesized by Dharmacon (Millenium Science). Non-targeting pool siRNA was used as siControl and purchased from Dharmacon (Millenium Science).

For overexpression experiments, the indicated cell lines were transfected with or 3-15 µg (for T25 and T75 flasks respectively) pCMV6-ALYREF-Myc/Flag (Origene Technologies, Rockville, MD, USA); pCMV6-Empty (Origene Technologies); pCMV3Tag3A-USP3^{wt}-Flag (GeneScript, Integrated Sciences, NSW, Australia); pCMV3Tag3A-USP3^{C168S}-Flag (GeneScript); pCMV3Tag3A-Empty (GeneScript); pCMV-His-Ubiquitin (GeneScript), pcDNA3.1-MYCN-HA or pcDNA3.1-Empty plasmid DNA. All HA-MYCN deletion mutants were generated in pcDNA3.1 vectors and provided by Professor Wei Gu. The catalytically defective enzyme mutant for USP3 with a serine residue substituting for cysteine (C168S) was generated by GeneScript from pCMV3Tag3A-USP3^{wt}-Flag by PCR-based site-directed mutagenesis. Cells were transfected between 24 to 96 hours depending on the

experimental requirements. Transfections were performed using Lipofectamine 2000 (Life Technologies) following the manufacturer's protocol.

Page 29

Immunoprecipitation

... HEK293T human embryonic cells were transiently co-transfected with a HA-MYCN-expression pcDNA3.1 construct, together with an empty vector or Flag-ALYREF-expression pCMV6 construct or Flag-USP3-expression pCMV3Tag3A construct for 24 hours. Cells were lysed in cold NP40 or BC100 buffer supplemented with protease inhibitor (Sigma), respectively. 5% of the cell extract was saved as the input, and the rest ...

Page 30

Western blot analysis

...K-63 linkage-specific ubiquitin (HWA4C4; 1:1000; Invitrogen, Thermo Fisher Scientific); ubiquitin (P4D1; 1:500; Cell Signaling Technologies); HA (3F10; 1:1000; Sigma); Topoisomerase I ...

Page 31

MYCN stability analysis

... Western blot to measure changes in MYCN protein stability.

HEK293T cells were co-transfected with pCMV3Tag3A-Flag-USP3^{WT} or pCMV3Tag3A-Flag-USP3^{C168S} mutant constructs along with pcDNA3.1-HA-MYCN plasmid DNA for 24 hours, then treated with 100 ug/ml cycloheximide (Sigma) for 0-60 minutes. Cells were collected at 15 minutes treatment intervals and protein lysates were analysed by Western blot to measure changes in MYCN protein stability.

Uncropped and unprocessed western blot images of the representative replicate used in main figures have been added to the Supplementary File, related to figures as indicated.

Figure 6

Fig. 6i

Fig. 6j

Fig. 6k

Fig. 6l

Figure 7

Fig.7a

Q7: The experiments showing co-recruitment of MYCN to the ALYREF binding site within the USP3 gene are not convincing (Figure 5e). First, they contradict published data for MYCN/RNAPII (Figure 5c) that show exclusive binding at the promoter, whereas ALYREF binds at a distal site around 6kb downstream. Second, the direct binding to the same genomic site may not be required for ALYREF to function as a cofactor for MYCN. The authors could examine available ChIP data for other transcription factors, chromatin conformation data, etc. Also, the authors could consider the possibility that ALYREF stabilizes MYCN via direct binding and steric effects (e.g., by interfering with a recruitment of a Ub ligase) rather than downstream transcriptional regulation - the ALYREF-MYCN binding data are strong, in contrast to the data for co-recruitment of ALYREF and MYCN to the USP3 locus.

Response: We conducted additional experiments to support our contention that despite the 6 kb distance between ALYREF and MYCN peaks at the USP3 gene (Original Figure 5d), the two proteins form a regulatory protein complex on the chromatin (Original Figure 5e). As ALYREF and MYCN binding sites do not overlap on the chromatin, we hypothesised that ALYREF-mediated chromatin alterations are required to enhance the transcriptional activation of USP3 by MYCN. We conducted chromatin conformation capture (3C) assay in SK-N-BE(2)C cells to further support this mechanism and the results are now described in the revised manuscript. Our data showed that the USP3 promoter sequence (TSS) containing the MYCN peak and the DNA sequence containing the ALYREF binding site (+6800 bp from TSS) are in

close proximity with each other. The DNA looping detected by the 3C assay was validated by using multiple controls and technical approaches. These new results along with the dual chromatin immunoprecipitation and gene expression experiments conducted in SHEPMYCN3 cells identify a previously unrecognised role of ALYREF – to promote regulatory chromatin interactions that support the malignant MYCN-driven neuroblastoma phenotype. The new chromatin conformation capture data has been added to Figure 5 and Supplementary Figure 5. Figure Legends, Supplementary Figure Legends, Results and Materials and Methods sections (containing all new method description and primer design) have been modified accordingly.

We have not ruled out the possibility that ALYREF may be stabilizing MYCN via a direct protein-protein interaction, and to investigate this mechanism is a natural extension to our study, however it is out of the scope of the current manuscript.

Changes in Figures:

Figure 5

Changes in Supplementary Figures:

Supplementary Figure 5

d

e

Changes in text:

Figure 5 legend

Page 49

...compared to negative control (Control). (f) Schematic of chromatin conformation capture (3C) assay used to detect the interactions between the DNA fragments containing the MYCN binding site at the USP3 promoter region (red, 0 bp) and that containing the ALYREF peak (green, + 6800bp from USP3 TSS). (g) Representative agarose gel from 3C assay performed with BamH1 and/or HindIII showing PCR product of MYCN and ALYREF interaction in SK-N-BE(2)C cells (PCR Amplicon #1; 166 bp) and the positive control non-cross-linked DNA (PCR Amplicon #2; 53 bp). The positions of molecular size standards (in base pairs) are shown to the right of the gel. (h) PCR analysis of USP3 expression...

Supplementary Figure 5 legend

...enrichment was compared to negative control (Control). (c) Schematic of 3C assay design at USP3 gene showing locations of BamH1 and HindIII restriction sites and DNA fragments containing the MYCN binding site at the USP3 promoter region (red, 0 bp) and that containing the ALYREF peak (green, + 6800bp from TSS). (d) Representative agarose gel from 3C assay showing PCR product of MYCN:ALYREF interaction in SK-N-BE(2)C cells (PCR Amplicon #1, lanes 1-8) and the positive control non-cross-linked DNA (PCR Amplicon #2, lanes 9-16). Genomic DNA, undigested genomic DNA from SK-N-BE(2)C cells; Genomic DNA +BamH1, BamH1 digested genomic DNA from SK-N-BE(2)C cells; Genomic DNA +BamH1 and HindIII, BamH1 and HindIII digested genomic DNA from SK-N-BE(2)C cells; Blank, no genomic DNA (water control); 3C product BamH1, product of PCR reaction following BamH1 digestion; 3C product BamH1 and Hind III, product of PCR reaction following BamH1 and HindIII digestion; 3C product BamH1 – ligase, ligase control for PCR reaction following BamH1 digestion; 3C product BamH1 and Hind III-ligase, ligase control for PCR reaction following BamH1 and HindIII digestion. The positions of molecular size standards (in base pairs) are shown to the left of the gel. (e) Electropherogram traces of a representative DNA Bioanalyzer run for 3C assay. FU=Fluorescence Unit; UM = Upper marker; LM = Lower marker peak. Arrows indicate the 3C PCR amplicon peaks. (f) Real-time PCR analysis of MYCN and ALYREF...

Results

Page 14

...in both Kelly and SK-N-BE(2)C cells, respectively (Fig. 5e). Binding of ALYREF to MYCN on chromatin might be due to a yet unknown function of ALYREF in regulating long-range interactions involving chromatin looping. To confirm the potential interaction between the USP3 promoter sequence binding MYCN at the TSS and the DNA sequence containing the ALYREF binding site at +6800 bp from the TSS, a chromatin conformation capture (3C) assay (Fig. 5f) was performed using restriction enzyme BamH1 (Supplementary Fig. 5c). A specific PCR product (PCR amplicon#1) was detected in BamH1 digested SK-N-BE(2)C samples (Fig. 5g and Supplementary Fig. 5d). Genomic DNA obtained from SK-N-BE(2)C cells without

cross-linking (Fig. 5f) was used as a positive control (PCR amplicon#2) in these experiments (Fig. 5f-g and Supplementary Fig. 5d). The DNA looping detected by the 3C assay was further validated using an additional restriction enzyme HindIII, with a restriction site in the ALYREF peak (Supplementary Fig. 5c). Restriction digestion with HindIII will prevent the ligation of the MYCN binding and ALYREF binding sequences, resulting in no PCR product using PCR amplicon #1 primers. As expected, when both HindIII and BamH1 were used, using the same PCR amplicon #1 primers pairs, no PCR product was obtained (Fig. 5g and Supplementary Fig. 5d). To confirm the 3C data and validate the assay, the PCR products were run on a DNA High Sensitivity (HS) chip using the Agilent 2100 Bioanalyzer (Supplementary Fig 5e). Amplified product was not detected from the sample in which both restriction enzymes were used but was present with BamH1 alone (Supplementary Fig 5e). These results suggest that the two DNA fragments are potentially interacting or are in close proximity with each other. To further examine ALYREF as a transcriptional coactivator for MYCN...

Materials and Methods

Page 33

Chromosome Conformation Capture (3C) assay

The 3C assay was carried out as previously described⁶⁷. Briefly, nuclei were extracted from fixed SK-N-BE(2)C cells as described in the chromatin immunoprecipitation protocol. BamH1 and HindIII (Promega Australia, NSW, Australia) restriction enzymes were selected based on *in silico* analysis to cut DNA. The digested nuclei were then treated with T4 DNA ligase (Thermo Fisher Scientific) to ligate DNA fragments that interacted with each other. DNA fragments after ligation were purified using phenol/chloroform/isoamyl alcohol. Then PCR was performed for 35 cycles using the following PCR conditions: 94°C for 10 min; and thermocycling at 94°C for 30 s, 60°C for 30 s and 72°C for 30 s; following by 72°C 15 min. PCR product was loaded on a 1.5% agarose gel with SYBR safe (Life Technologies), and images were taken using a ChemiDoc Imaging System (Bio-Rad). PCR samples from 3C assay were also run on a DNA High Sensitivity (HS) chip using the Agilent 2100 Bioanalyzer (Agilent Technologies Australia, VIC, Australia) according to the manufacturer's instructions.

Uncropped and unprocessed PCR gel image of the representative replicate used in main figures have been added to the Supplementary File, related to figures as indicated.

Fig. 5g

Q8. The authors should proofread the manuscript and rephrase sentences which are grammatically incorrect or scientifically inaccurate. For example, line 337: “USP3 expression associates with MYCN-amplification and is selectively dependent for neuroblastoma cell viability”. Line 417: “Transcriptional regulators undergo rapid and timely degradation by ubiquitination”.

Authors Z. Nagy, B.B. Cheung and G.M. Marshall have proofread the manuscript.

REVIEWERS' COMMENTS

Reviewer #1 (Remarks to the Author):

I'm happy with the adjustments and extra data provided by the authors

Reviewer #2 (Remarks to the Author):

In the revised version of the manuscript "An ALYREF-MYCN coactivator complex drives neuroblastoma tumorigenesis through effects on USP3 and MYCN stability", Marshall and colleagues have addressed all of the points raised during the review of the original manuscript.

The authors have addressed the points regarding the methods used (Alamar Blue staining and BrdU incorporation assays) and corroborated previous results with colony formation assays (Figure 1, Figure 7).

A more detailed description of the immunoprecipitation assay used to determine the effects on MYCN ubiquitination (Figure 3, Figure 7) is provided now in the Materials and Methods section.

The authors have clarified the proposed mechanism of the MYCN-ALYREF cross-regulation in the text and have addressed it with additional experiments, including the chromatin conformation capture assays (Figure 5).

The authors have now used the catalytically inactive variant of USP3 to demonstrate the requirement of the deubiquitinase activity in MYCN stabilization (Figure 6).

Collectively, I believe that these new experiments and updated text have significantly clarified and strengthened the manuscript. I have no further concerns with the current dataset.

Manuscript Number: NCOMMS-20-19257A

Manuscript Title: An ALYREF-MYCN coactivator complex drives neuroblastoma tumorigenesis through effects on USP3 and MYCN stability

Re: Submission of the 2nd revised manuscript

We would like to thank the editorial board and the reviewers for reviewing the revised manuscript. We appreciate the favourable evaluation and comments. The new experiments and data have strengthened the manuscript. A point by point response to the reviewer's comments have been prepared as per following.

Key for responses

Reviewer Comments to the Author– Italics

Author Responses – Normal black text

Reviewer #1

Q1: I'm happy with the adjustments and extra data provided by the authors

Response: Thank you for reviewing the revised manuscript. No revisions are required.

Reviewer #2

Q1: In the revised version of the manuscript “An ALYREF-MYCN coactivator complex drives neuroblastoma tumorigenesis through effects on USP3 and MYCN stability”, Marshall and colleagues have addressed all of the points raised during the review of the original manuscript.

The authors have addressed the points regarding the methods used (Alamar Blue staining and BrdU incorporation assays) and corroborated previous results with colony formation assays (Figure 1, Figure 7).

A more detailed description of the immunoprecipitation assay used to determine the effects on MYCN ubiquitination (Figure 3, Figure 7) is provided now in the Materials and Methods section.

The authors have clarified the proposed mechanism of the MYCN-ALYREF cross-regulation in the text and have addressed it with additional experiments, including the chromatin conformation capture assays (Figure 5).

The authors have now used the catalytically inactive variant of USP3 to demonstrate the requirement of the deubiquitinase activity in MYCN stabilization (Figure 6).

Collectively, I believe that these new experiments and updated text have significantly clarified and strengthened the manuscript. I have no further concerns with the current dataset.

Response: Thank you for reviewing the revised manuscript. No revisions are required.